# Online Learning with Sublinear Best-Action Queries

**Matteo Russo**
Sapienza University of Rome, Italy
`mrusso@diag.uniroma1.it`

**Andrea Celli**
Bocconi University, Italy
`andrea.celli2@unibocconi.it`

**Riccardo Colini-Baldeschi**
Meta, Central Applied Science, UK
`rickuz@meta.com`

**Federico Fusco**
Sapienza University of Rome, Italy
`fuscof@diag.uniroma1.it`

**Daniel Haimovich**
Meta, Central Applied Science, UK
`danielha@meta.com`

**Dima Karamshuk**
Meta, Central Applied Science, UK
`karamshuk@meta.com`

**Stefano Leonardi**
Sapienza University of Rome, Italy
`leonardi@diag.uniroma1.it`

**Niek Tax**
Meta, Central Applied Science, UK
`niek@meta.com`

## Abstract

In online learning, a decision maker repeatedly selects one of a set of actions, with the goal of minimizing the overall loss incurred. Following the recent line of research on algorithms endowed with additional predictive features, we revisit this problem by allowing the decision maker to acquire additional information on the actions to be selected. In particular, we study the power of *best-action queries*, which reveal beforehand the identity of the best action at a given time step. In practice, predictive features may be expensive, so we allow the decision maker to issue at most $k$ such queries. We establish tight bounds on the performance any algorithm can achieve when given access to $k$ best-action queries for different types of feedback models. In particular, we prove that in the full feedback model, $k$ queries are enough to achieve an optimal regret of $\Theta(\min\{\sqrt{T}, T/k\})$. This finding highlights the significant multiplicative advantage in the regret rate achievable with even a modest (sublinear) number $k \in \Omega(\sqrt{T})$ of queries. Additionally, we study the challenging setting in which the only available feedback is obtained during the time steps corresponding to the $k$ best-action queries. There, we provide a tight regret rate of $\Theta(\min\{T/\sqrt{k}, T^2/k^2\})$, which improves over the standard $\Theta(T/\sqrt{k})$ regret rate for label efficient prediction for $k \in \Omega(T^{2/3})$.

## 1 Introduction

Online learning is a foundational problem in machine learning. In its simplest version, a decision maker repeatedly interacts with a fixed set of $n$ actions over a time horizon $T$. At each time, the decision maker needs to choose one of a set of actions; subsequently, it receives an action-dependent loss and observes some feedback. These loss functions are generated by an omniscient (but oblivious) adversary and are only revealed on-the-go. The goal of the decision maker is to design a learning algorithm that achieves small *regret* with respect to the best fixed action in hindsight, i.e., the difference between the decision maker's loss and that of the fixed action. Several online learning algorithms

have been developed, characterized by optimal instance-independent regret bound, depending on the feedback model [Cesa-Bianchi and Lugosi, 2006, Slivkins, 2019].

Following the recent literature on algorithms with machine learning-based predictions (see, e.g., the survey by Mitzenmacher and Vassilvitskii [2020]), we study the case where the learner is allowed to issue a limited number of *best-action queries* to an oracle that reveals the identity of the best action for that step, so that the learner can choose it. This setting is motivated by scenarios in which obtaining accurate predictions on the optimal choice among numerous actions is possible but comes with significant costs and time constraints. For instance, consider an online platform that continuously moderates posted content (e.g., Meta [Meta, a,b] or Google [Google]), and the online learning problem it faces: posts are generated one after the other, and the platform's task consists in deciding whether or not to flag the content as harmful. In this application, the platform may do so via (i) content moderation actions that are based on (expert) human reviews (that plays the role of *best-action queries*), and (ii) automated content moderation decisions, i.e., decisions arising by employing an online learning algorithm. Due to budget or time constraints, access to human reviewing is a scarce resource, and the platform can only employ external reviewers at most $k$ times.

While the idea of incorporating hints or queries into online learning models has already been studied [e.g., Bhaskara et al., 2021b, 2023b], we are the first to study *best-action queries*. Bhaskara et al. [2021b] focus on online linear optimization with full feedback, with a query model which outputs vectors that are correlated with the actual losses. In the case of optimization over strongly convex domains, the regret bound improves from $\sqrt{T}$ to $\log T$, even for learners that receive hints for $O(\sqrt{T})$ times. In an alternative model, Bhaskara et al. [2023b] studies comparison queries that allow the decision maker to know in advance, at each time, which among a small number of actions has the best loss. In this model, probing 2 arms is sufficient to achieve time-independent regret bounds for online linear and convex optimization with full feedback, an in the stochastich multi-armed bandit problem. Our model differs from previous ones in two directions: (i) the online learner issues at most $k$ queries (differently from Bhaskara et al. [2023b]), and (ii) these queries are purely ordinal and the domain is not strongly convex[*] (so that the logarithmic bound in Bhaskara et al. [2021b] does not apply).

Another potential way of addressing how to efficiently using limited moderation is through the abstention learning model, where the learner can "abstain" from making a decision and instead use additional resources to receive the optimal response. This includes models like KWIK and full-information models [Li et al., 2011, Sayedi et al., 2010, Zhang and Chaudhuri, 2016, Cortes et al., 2018, Neu and Zhivotovskiy, 2020, Gangrade et al., 2021].

## 1.1 Our Model

In our model, an online learner repeatedly interacts with $n$ actions over a time horizon $T$. At the beginning of each time $t \in [T]$[†], the learner chooses one of these actions $i_t$ and suffers a loss $\ell_t(i_t)$ generated by an (oblivious) adversary that may depend on both the action and the time; then, it observes some feedback. In this paper, we allow the learning algorithm $\mathcal{A}$ to issue a *best-action query*, for at most $k$ out of $T$ times. When the learner issues a query, an oracle reveals the *identity* of the best action at that time, $i_t^*$, so that the learner can select it. The quality of a learning algorithm is measured via the regret: the difference between its performance and that of the best fixed action in hindsight. The regret of an algorithm $\mathcal{A}$ against a sequence of losses $\boldsymbol{\ell} \in [0,1]^{n \times T}$ reads

$$R_T(\mathcal{A}, \boldsymbol{\ell}) = \sum_{t \in [T]} \mathbb{E}\left[\ell_t(i_t)\right] - \min_{i \in [n]} \sum_{t \in [T]} \ell_t(i),$$

where the expectation runs over the (possibly) randomized decisions of the algorithm. We are interested in designing learning algorithms that perform well, i.e., suffer sublinear regret against all possible sequences of losses. For this reason, we denote with $R_T(\mathcal{A})$ (without the dependence on $\boldsymbol{\ell}$) the worst-case regret of $\mathcal{A}$: $R_T(\mathcal{A}) = \sup_{\boldsymbol{\ell}} R_T(\mathcal{A}, \boldsymbol{\ell})$. The minimax regret of a learning problem is then the regret achievable by the best algorithm. In our paper we pinpoint the exact minimax regret rates for the problems studied.

---

[*]The model with $k$ actions can be captured by a linear function where each entry corresponds to the loss of a specific discrete action, and the continuous action space is the probability simplex over the $n$ discrete actions (which is not strongly convex).

[†]We adopt the notational convention that $[x]$ stands for the set of the first $x$ natural numbers.

For the sake of simplicity, we denote with $L_T(i)$ the total loss incurred by action $i$ over the whole time horizon: $L_T(i) = \sum_{t \in [T]} \ell_t(i)$. $L_T^{\min} = \sum_{t \in [T]} \ell_t(i_t^*)$ denotes instead the sum of the minimum loss actions at each time. Finally, $L_T(\mathcal{A}_k) = \mathbb{E}\left[\sum_{t \in [T]} \ell_t(i_t)\right]$ denotes the expected total loss of an algorithm $\mathcal{A}_k$ issuing at most $k$ best-action queries.

## 1.2 Our Results

**Full feedback.**   We start with the *full feedback* (a.k.a. prediction with experts) where the learner observes at each time the losses of all the actions after the action is chosen, i.e., *all* the loss vector $\boldsymbol{\ell}_t = (\ell_t(1), \ell_t(2), \ldots, \ell_t(n))$ after action $i_t$ is chosen. We obtain the following results:

- We show that by combining the Hedge algorithm with $k$ queries issued uniformly at random (Algorithm 1), we obtain a regret rate of $O(\min\{\sqrt{T}, T/k\})$, see Theorem 2.2.
- In Theorem 3.3, we complement this positive result with a tight lower bound: we design a family of instances that forces every algorithm with $k$ queries to suffer the same regret.

It is surprising that issuing a sublinear number of queries, say $k = T^\alpha$ for $\alpha \in (1/2, 1)$, yields a significant improvement on the regret, which becomes $T^{1-\alpha}$. Note, the total of the losses incurred by any algorithm in $k$ rounds is at most $k$, which only affects the overall regret in an additive way; nevertheless, we prove that issuing $k$ queries has an impact on the regret that is *multiplicative* in $k$. For instance, $T^{2/3}$ queries are enough to decrease the regret to $T^{1/3}$, which is well below the $\Theta(\sqrt{T})$ minimax regret for prediction with expert *without* queries [Cesa-Bianchi and Lugosi, 2006].

**Label efficient feedback.**   We then proceed to study a partial feedback model inspired by the label efficient paradigm [Cesa-Bianchi and Lugosi, 2006]. In the label-efficient prediction problem, the learner only observes the losses in a few selected times. With *label efficient feedback,* the learner only observes feedback during the $k$ times where the best-action queries are issued (but only after choosing the action). We obtain the following results:

- We modify the (label-efficient) Hedge algorithm [Cesa-Bianchi and Lugosi, 2006] to achieve a regret rate of $O(\min\{T/\sqrt{k}, T^2/k^2\})$, see Theorem 2.4,
- In Theorem 3.4, we show that it is not possible to improve on these rates: there exists instances where all algorithms suffer $\Omega(T/\sqrt{k})$ regret if $k \in O(T^{2/3})$ or $\Omega(T^2/k^2)$ if $k \in \Omega(T^{2/3})$.

We observe that our algorithms improve (for $k \in \Omega(T^{2/3})$) on the regret rate of label efficient prediction with $k$ steps of feedback, which is of order $\Theta(T/\sqrt{k})$ [Chapter 6.2 in Cesa-Bianchi and Lugosi, 2006]. Also in this case, we observe the surprising multiplicative impact on the regret of the $k$ queries. For instance, $k = T^{3/4}$ queries (which impact on a total loss of the same order but leave unaffected $\Theta(T)$ rounds) are enough to achieve $O(\sqrt{T})$ regret.

**Stochastic i.i.d. setting.**   In Section B, we derive (up to polylogarithmic factors) the above results in the stochastic setting, but with different algorithms. Namely, in full feedback, we show how Follow-The-Leader [Auer et al., 2003] achieves regret $\Theta(T/k)$ when $k \geq \Omega(\sqrt{T})$, and $\tilde{\Theta}(\sqrt{T})$ otherwise. With label efficient feedback, Explore-Then-Commit [Perchet et al., 2015] achieves $\tilde{\Theta}(T^2/k^2)$ when $k \geq \Omega(T^{2/3})$, and $\tilde{\Theta}(T/\sqrt{k})$ otherwise.[‡]

## 1.3 Technical Challenges

Issuing a best-action query has the effect of inducing a possibly *negative* instantaneous regret. In fact, the benchmark that the algorithm is comparing against at time $t$ is the loss of the best fixed action over the whole time horizon, which may naturally be larger, at time $t$, than the best action $i_t^*$ in that specific time. Overall, the magnitude of the total "negative" regret that is generated by $k$ queries is at most $O(k)$, which is typically sublinear in $T$, and affects *linearly* on the overall regret. We prove that this sublinear number of queries has a (surprising) multiplicative effect on the overall regret bound.

---

[‡]We use $\tilde{\Theta}(x)$ to hide polylogarithmic terms in $x$.

Our analysis reveals that any algorithm employing a *uniform querying* strategy can be characterized by a total loss decomposed into two terms: a $k/T$ fraction of the best dynamic loss, representing the sum of the smallest losses at each time, and the total loss of the same algorithm operating without queries, scaled by a factor of $1 - k/T$. Additionally, within our proof, we conduct a more refined analysis of the Hedge algorithm in the *no-querying* setting. Specifically, we can express the regret as a function of the difference between the best dynamic loss and the best-fixed cumulative loss in hindsight. This formulation demonstrates a noteworthy enhancement in regret rates compared to the conventional bound of $O(\sqrt{T})$ when the discrepancy between the best dynamic loss and the best-fixed cumulative loss in hindsight is relatively small.

Concerning lower bounds, their construction is more challenging than their "no queries" counterpart. In fact, we need to design hard instances against *any* query-augmented learning algorithm, and provide tight bounds in both $k$ and $T$. The lower bounds we construct are stochastic, this implies that the minimax regret we find are tight, even in the stochastic model, where the losses are drawn i.i.d. from a fixed but unknown distribution. To exemplify this, let us consider the following natural instance, which provides a simple proof of the $\Omega(\sqrt{T})$ regret lower bound in the adversarial setting (without queries). The instance is composed of two arms, whose rewards are i.i.d. Bernoulli distribution with probability $1/2$. Any learning algorithm achieves $T/2$ regret, while the best-fixed arm in hindsight is expected to achieve an extra $\Theta(\sqrt{T})$ term.[§] Now, if the learner is given the power to issue (order of) $\sqrt{T}$ queries, then its regret naturally drops from (order of) $\sqrt{T}$ to constant.

## 2 Hedge with $k$ Best-Action Queries

In this section, we propose two algorithms that address online learning with full and label efficient feedback. They are built by combining the Hedge algorithm [Chapter 2 in Cesa-Bianchi and Lugosi, 2006] to uniform queries. We start by presenting some known properties of Hedge (in full feedback) that are crucial key for the main results of this section.

**Lemma 2.1.** *Consider the Hedge algorithm* $\text{Hedge}_\eta(\tilde{\ell})$ *run on loss sequence* $\tilde{\ell} \in [0, U]^{n \times T}$ *with learning rate* $\eta < 1/U$. *Then, for all action* $i \in [n]$, *it holds that,*

$$\tilde{L}_T(\text{Hedge}_\eta(\tilde{\ell})) \leq \frac{1}{1 - U\eta} \cdot \left( \tilde{L}_T(i) + \frac{\log n}{\eta} \right),$$

*where* $\tilde{L}_T(\text{Hedge}_\eta(\tilde{\ell})) = \sum_{t \in [T], i \in [n]} p_t(i) \cdot \tilde{\ell}_t(i)$ *is the expected cumulative loss of* $\text{Hedge}_\eta(\tilde{\ell})$ *and* $\tilde{L}_T(i)$ *is the cumulative loss of action* $i$.

*Proof.* We have that, by definition,

$$W_{T+1} \geq w_{T+1}(i) w_t(i) \cdot \exp\left(-\eta \tilde{\ell}_t(i)\right) = \prod_{t \in [T]} \exp\left(-\eta \tilde{\ell}_t(i)\right) = \exp\left(-\eta \tilde{L}_T(i)\right).$$

We also know that

$$W_{t+1} \leq W_t \cdot \left( 1 - \eta \sum_{i \in [n]} p_t(i) \cdot \tilde{\ell}_t(i) + \eta^2 \sum_{i \in [n]} p_t(i) \cdot \tilde{\ell}_t^2(i) \right).$$

Thus,

$$W_{T+1} \leq n \cdot \prod_{t \in [T]} \left( 1 - \eta \sum_{i \in [n]} p_t(i) \cdot \tilde{\ell}_t(i) + \eta^2 \sum_{i \in [n]} p_t(i) \cdot \tilde{\ell}_t^2(i) \right),$$

which combined with the earlier bound and taking logarithms of both sides (since they are both positive), gives

$$-\eta \tilde{L}_T(i) \leq \log n + \sum_{t \in [T]} \log \left( 1 - \eta \sum_{i \in [n]} p_t(i) \cdot \tilde{\ell}_t(i) + \eta^2 \sum_{i \in [n]} p_t(i) \cdot \tilde{\ell}_t^2(i) \right)$$

---

[§]This is a simple corollary of the expected distance of a random walk.

$$\leq \log n - \eta \sum_{t\in[T],i\in[n]} p_t(i)\cdot\tilde{\ell}_t(i) + \eta^2 \sum_{t\in[T],i\in[n]} p_t(i)\cdot\tilde{\ell}_t^2(i)$$

$$\leq \log n - (\eta - U\eta^2) \sum_{t\in[T],i\in[n]} p_t(i)\cdot\tilde{\ell}_t(i).$$

The second inequality above follows from $\log(1+z)\leq z$ for $z\in\mathbb{R}$, and the third by observing that $\tilde{\ell}_t^2(i)\leq U\tilde{\ell}_t(i)$, since $\tilde{\ell}_t(i)\in[0,U]$. The lemma follows by rearranging the terms above. $\qquad\square$

## 2.1 Full Feedback: An $O(\frac{T\log n}{k})$ Regret Bound

**Theorem 2.2.** *Consider the problem of online learning with full feedback and $k$ best-action queries, then there exists an algorithm $\mathcal{A}_k$ that guarantees*

$$R_T(\mathcal{A}_k) \leq \min\left\{\sqrt{T\log n}, \frac{T\log n}{k}\right\}.$$

We prove that Algorithm 1 exhibits the desired regret guarantees. In particular, as we illustrate next, this algorithm performs uniform querying, i.e., they choose a uniformly random subset $Q\subseteq[T]$ of size $k$ where to issue best-action queries. In the case of uniform queries, and when feedback and action taken by the algorithm are not correlated, a useful simplification can be made.

**Observation 2.3.** *Let $\mathcal{A}_0$ be an algorithm with no querying power, with full or label-efficient feedback. Consider $\mathcal{A}_k$, obtained from $\mathcal{A}_0$ by equipping it with $k$ uniformly random queries across the time horizon $T$ or with an independent query on each step time step with probability $k/T$. Similarly, let $i_t^0$ and $i_t$ be the actions selected by $\mathcal{A}_0$ and $\mathcal{A}_k$ at time $t$. Then, for all adversarial sequences $\boldsymbol{\ell}\in[0,1]^{n\times T}$ of action losses,*

$$\mathbb{E}\left[\ell_t(i_t)\right] = \left(1-\frac{k}{T}\right)\cdot\mathbb{E}\left[\ell_t(i_t^0)\right] + \frac{k}{T}\cdot\mathbb{E}\left[\ell_t(i_t^*)\right],$$

*for all $t\in[T]$, and thus*

$$L_T(\mathcal{A}_k) = \left(1-\frac{k}{T}\right)\cdot L_T(\mathcal{A}_0) + \frac{k}{T}\cdot L_T^{\min}. \tag{1}$$

---

**Algorithm 1** Hedge with Best-Action Queries

---

1: **Input:** Sequence of losses $\ell_t(i)$ and query budget $k\in[T]$
2: Sample $k$ out of $T$ time steps uniformly at random and denote this random set by $Q$
3: Set $\eta = \sqrt{\frac{\log n}{T}}$ when $k\leq\sqrt{T}$, otherwise $\eta = \frac{k}{T}$
4: Initialize $w_1(i) = 1$ for all $i\in[n]$
5: **for** $t\in[T]$ **do**
6:     **if** $t\in Q$ **then**
7:         Observe $i_t^* = \arg\min_{i\in[n]}\ell_t(i)$
8:         Select action $i_t^*$
9:     **else**
10:         Let $W_t = \sum_{i\in[n]} w_t(i)$
11:         Select action $i$ with probability $p_t(i) = \frac{w_t(i)}{W_t}$
12:     Observe $\ell_t(i)$ for all $i\in[n]$
13:     Update $w_{t+1}(i) = w_t(i)\cdot\exp\left(-\eta(\ell_t(i)-\ell_t(i_t^*))\right)$ for all $i\in[n]$

---

*Proof of Theorem 2.2.* Let us first note that Algorithm 1 without queries is an instantiation $\text{Hedge}_\eta(\tilde{\boldsymbol{\ell}})$ applied to losses $\tilde{\ell}_t(i) = \ell_t(i)-\ell_t(i_t^*)\in[0,1]$. Let this algorithm be denoted as $\mathcal{A}_0$. Then, applying Lemma 2.1 and expanding the terms, we obtain

$$L_T(\mathcal{A}_0) \leq \frac{L_T(i)}{1-\eta} + \frac{\log n}{\eta(1-\eta)} - \frac{\eta L_T^{\min}}{1-\eta}.$$

Let $\mathcal{A}_k$ be Algorithm 1 with $k$ best-action queries. By Observation 2.3, specifically (1), it holds that

$$L_T(\mathcal{A}_k) \leq \frac{(1 - {k}/{T})L_T(i)}{1 - \eta} + \frac{(1 - {k}/{T})\log n}{\eta(1 - \eta)} - \frac{\eta(1 - {k}/{T})L_T^{\min}}{1 - \eta} + \frac{k}{T} \cdot L_T^{\min}$$

$$\iff R_T(\mathcal{A}_k, i) \leq \frac{(1 - {k}/{T})\log n}{\eta(1 - \eta)} + \frac{T\eta - k}{T(1 - \eta)}(L_T(i) - L_T^{\min}) \leq \min\left\{\sqrt{T \log n}, \frac{T \log n}{k}\right\},$$

where the last inequality holds by setting $\eta = \max\left\{\sqrt{{\log n}/{T}}, {k}/{T}\right\}$. $\qquad \square$

Before proceeding further, let us provide some technical intuition of why Theorem 2.2 should hold. The additive term $\frac{k}{T} \cdot L_T^{\min}$ (given in Equation (1)) alters the choice of the learning rate, which is allowed to be more aggressive, thus impacting the regret in a multiplicative way. To be more specific, in the usual Hedge performance analysis, we do not care about the negative term $\frac{-\eta L_T^{\min}}{1 - \eta}$. This negative term together with the additive $\frac{k}{T} \cdot L_T^{\min}$ term given by best-action queries allows us to set the optimal $\eta$ to be larger than the usual (order of) $1/\sqrt{T}$. In other words, the additive impact of the $\frac{k}{T} \cdot L_T^{\min}$ term permits a multiplicative gain in regret as the learning rate $\eta$ is modified and increased.

## 2.2 Label Efficient Feedback: An $O(\frac{T^2 \log n}{k^2})$ Regret Bound

We extend Algorithm 1 to a setting where feedback is given only during querying time steps, with the only difference that the update rule is performed just after the querying time steps and nowhere else across the time horizon. We prove the following theorem:

**Theorem 2.4.** *For all adversarial sequences $\boldsymbol{\ell} \in [0,1]^{n \times T}$ of action losses and for all $k \geq \sqrt{\frac{T \log T}{2}} - 1$, in the label efficient query model, there exists an algorithm $\mathcal{A}_k$ that guarantees*

$$R_T(\mathcal{A}_k) \leq 2 \cdot \min\left\{T\sqrt{\frac{2 \log n}{k}}, \frac{T^2 \log n}{k^2}\right\}.$$

**Algorithm description.** We first describe the algorithm $\mathcal{A}_k$ we use: Let $X_t \sim \text{Ber}\left(\hat{k}/T\right)$ be a Bernoulli random variable, for some $\hat{k} \leq k$ to be specified later. $\mathcal{A}_k$ issues a best action query if, at time step $t$, $X_t = 1$ and unless the query budget is exhausted. Once the query budget is exhausted, the algorithm stops querying. Otherwise, it performs the usual update rule on losses $\hat{\ell}_t(i) = \frac{T}{k} \cdot (\ell_t(i) - \ell_t(i_t^*)) \cdot \mathbb{I}\{X_t = 1\}$. The algorithm then simply selects action $I_t = i_t^*$ if $X_t = 1$ and action $I_t = i$ with probability $p_t(i)$ if $X_t = 0$. Moreover, we denote by $X_{\leq t} = (X_1, \dots, X_t), I_{\leq t} = (I_1, \dots, I_t)$.

For the sake of the analysis, we introduce another algorithm $\mathcal{A}_k'$, which is the same as algorithm $\mathcal{A}_k$ with the only (but crucial) difference that it issues a query if and only if $X_t = 1$, regardless of whether or not query budget is exhausted. We thus bound the regret of $\mathcal{A}_k$ in terms of the regret of $\mathcal{A}_k'$.

**Lemma 2.5.** *For all adversarial sequences $\boldsymbol{\ell} \in [0,1]^{n \times T}$ of action losses, in the label efficient query model, algorithm $\mathcal{A}_k'$ guarantees*

$$R_T(\mathcal{A}_k') \leq \min\left\{T\sqrt{\frac{2 \log n}{\hat{k}}}, \frac{T^2 \log n}{\hat{k}^2}\right\}.$$

*Proof.* For algorithm $\mathcal{A}_k'$, we have that its counterpart without queries, $\mathcal{A}_0'$, is an instantiation $\text{Hedge}_\eta(\tilde{\boldsymbol{\ell}})$, with $\tilde{\boldsymbol{\ell}} = \hat{\boldsymbol{\ell}}$. Thus, by Lemma 2.1, we obtain

$$\hat{L}_T(\mathcal{A}_k') = \sum_{t \in [T], i \in [n]} p_t(i) \cdot \hat{\ell}_t(i) \leq \frac{1}{1 - \frac{T}{\hat{k}}\eta} \cdot \left(\hat{L}_T(i) + \frac{\log n}{\eta}\right), \qquad (2)$$

as long as $\eta < \hat{k}/T$. We now recognize that, since $\mathbb{E}\left[\hat{\ell}_t(i) \mid X_{\leq t-1}, I_{\leq t-1}\right] = \ell_t(i) - \ell_t(i_t^*)$, then

$$\sum_{i \in [n]} \mathbb{E}\left[p_t(i) \cdot \hat{\ell}_t(i) \Big| X_{\leq t-1}, I_{\leq t-1}\right] = \sum_{i \in [n]} p_t(i) \cdot (\ell_t(i) - \ell_t(i_t^*)).$$

Therefore, by the tower property of expectation applied around (2), we have

$$L_T(\mathcal{A}_0') = \sum_{i \in [n]} \mathbb{E}\left[p_t(i) \cdot \hat{\ell}_t(i)\right] = \sum_{i \in [n]} p_t(i) \cdot (\ell_t(i) - \ell_t(i_t^*))$$

$$\leq \frac{1}{1 - \frac{T}{\hat{k}}\eta} \cdot \left(L_T(i) - L_T^{\min} + \frac{\log n}{\eta}\right) \leq \frac{\hat{k}}{\hat{k} - T\eta} \cdot \left(L_T(i) + \frac{\log n}{\eta}\right) - \frac{T\eta}{\hat{k} - T\eta} L_T^{\min},$$

where the last inequality holds since $T\eta \leq \hat{k}$. By Observation 2.3, we get

$$L_T(\mathcal{A}_k') \leq \left(1 - \frac{k}{T}\right) \cdot \frac{\hat{k}}{\hat{k} - T\eta} \cdot \left(L_T(i) + \frac{\log n}{\eta}\right) - \frac{(T-k)\eta}{\hat{k} - T\eta} L_T^{\min} + \frac{k}{T} L_T^{\min},$$

which means that the regret is upper bounded by

$$R_T(\mathcal{A}_k', i) \leq \frac{\hat{k} \log n}{\eta(\hat{k} - T\eta)} + \frac{T^2\eta - k\hat{k}}{T(\hat{k} - T\eta)}(L_T(i) - L_T^{\min}) \leq \min\left\{T\sqrt{\frac{2\log n}{\hat{k}}}, \frac{T^2 \log n}{\hat{k}^2}\right\},$$

where the last inequality holds by setting $\eta = \max\left\{\frac{1}{T}\sqrt{\frac{\hat{k}\log n}{2}}, \frac{k\hat{k}}{\sqrt{2}T^2}\right\}$, and then by noticing, in the latter case, that $\hat{k} - \frac{k\hat{k}}{\sqrt{2}T} \geq \sqrt{2}k$. $\qquad \square$

With this lemma, we are ready to prove Theorem 2.4, where we bound the regret of algorithm $\mathcal{A}_k$.

*Proof of Theorem 2.4.* We consider event $\mathcal{E} = \{|Q| \leq k\}$, so that, slightly abusing notation, we write the regret of algorithm $\mathcal{A}_k$ as

$$R_T(\mathcal{A}_k) = \mathbb{E}\left[R_T(\mathcal{A}_k) \mid \mathcal{E}\right] \cdot \mathbb{P}\left[\mathcal{E}\right] + \mathbb{E}\left[R_T(\mathcal{A}_k) \mid \bar{\mathcal{E}}\right] \cdot \mathbb{P}\left[\bar{\mathcal{E}}\right] \leq \mathbb{E}\left[R_T(\mathcal{A}_k) \mid \mathcal{E}\right] + T \cdot \mathbb{P}\left[\bar{\mathcal{E}}\right].$$

To upper bound the second summand above, we have

$$T \cdot \mathbb{P}\left[\bar{\mathcal{E}}\right] \leq T \cdot \exp\left(-\frac{2(k + 1 - \hat{k})^2}{T}\right) \leq T \cdot \frac{1}{T} = 1, \tag{3}$$

by Hoeffding's inequality applied on the binomial random variable $|Q|$ with expectation $\hat{k}$, and as long as $\hat{k} \geq k - \sqrt{\frac{T \log T}{2}} + 1$.

For what concerns the first summand, we recognize that under event $\mathcal{E}$, $\mathcal{A}_k$ and $\mathcal{A}_k'$ are exactly the same algorithm. Thus, it holds that $\mathbb{E}\left[R_T(\mathcal{A}_k, i) \mid \mathcal{E}\right] = \mathbb{E}\left[R_T(\mathcal{A}_k', i) \mid \mathcal{E}\right]$. Moreover, if $\mathbb{E}\left[R_T(\mathcal{A}_k) \mid \bar{\mathcal{E}}\right] \geq 0$, and since $\mathbb{P}\left[\mathcal{E}\right] \geq 1 - 1/T$, then

$$\mathbb{E}\left[R_T(\mathcal{A}_k', i) \mid \mathcal{E}\right] \leq \frac{T}{T-1} \cdot R_T(\mathcal{A}_k', i) \leq \frac{T}{T-1} \cdot \min\left\{T\sqrt{\frac{2\log n}{\hat{k}}}, \frac{T^2 \log n}{\hat{k}^2}\right\},$$

by Lemma 2.5. If, instead, $\mathbb{E}\left[R_T(\mathcal{A}_k) \mid \bar{\mathcal{E}}\right] < 0$, we also know that, by an identical derivation to (3), $\mathbb{E}\left[R_T(\mathcal{A}_k) \mid \bar{\mathcal{E}}\right] \cdot \mathbb{P}\left[\bar{\mathcal{E}}\right] \geq -1$. Therefore,

$$\mathbb{E}\left[R_T(\mathcal{A}_k', i) \mid \mathcal{E}\right] \leq \frac{T}{T-1} \cdot (R_T(\mathcal{A}_k', i) + 1) \leq \frac{T}{T-1} \cdot \left(\min\left\{T\sqrt{\frac{2\log n}{\hat{k}}}, \frac{T^2 \log n}{\hat{k}^2}\right\} + 1\right),$$

again by Lemma 2.5. Overall, we obtain

$$R_T(\mathcal{A}_k) \leq 2 \cdot \min\left\{T\sqrt{\frac{2\log n}{k}}, \frac{T^2 \log n}{k^2}\right\}. \qquad \square$$

# 3 Lower Bounds

In this section, we construct two of randomized instances of the learning problem, which induce a tight lower bound on the minimax regret rates for both feedback models. We define the random variables $Z_t$ as the feedback observed by the algorithm at the end of time $t$. In the full feedback model, $Z_t = \ell_t$, while in the label efficient setting, $Z_t = \ell_t$ only if a query is issued at time $t$ (and $Z_t$ is an empty $n$-dimensional vector otherwise). Furthermore, we denote with $Z_{\leq t}$ the array containing the feedback $Z_1, Z_2, \ldots, Z_t$ until time $t$.

We start by describing the two stochastic instances, which are characterized by two distributions over two losses. As a notational convention, we denote the losses with $\ell_t$, and introduce two probability measures $\mathbb{P}^+, \mathbb{P}^-$ (and their corresponding expectations $\mathbb{E}^+, \mathbb{E}^-$). Let $\varepsilon, q \in [0, 1]$ be two parameters we set later (with $\varepsilon \leq q$), we have that the losses of the $n = 2$ actions are distributed as follows:

$$(\ell_t(1), \ell_t(2)) = \begin{cases} (1,1) & \text{w.p. } \frac{1}{2} \text{ under both } \mathbb{P}^+ \text{ and } \mathbb{P}^- \\ (0,0) & \text{w.p. } \frac{1}{2} - 2q \text{ under both } \mathbb{P}^+ \text{ and } \mathbb{P}^- \\ (0,1) & \text{w.p. } q + \varepsilon \text{ under } \mathbb{P}^+ \text{ and w.p. } q - \varepsilon \text{ under } \mathbb{P}^- \\ (1,0) & \text{w.p. } q - \varepsilon \text{ under } \mathbb{P}^+ \text{ and w.p. } q + \varepsilon \text{ under } \mathbb{P}^- \end{cases}.$$

We now introduce and prove a general Lemma on the expected regret $R_T^\pm(\mathcal{A}_k)$ suffered by any deterministic algorithm $\mathcal{A}_k$ which issues at most $k$ queries, against the i.i.d. sequence of valuations drawn according to $\mathbb{P}^\pm$. Since we want a result that holds for both feedback models, we introduce the random set $F$ which contains the times where $\mathcal{A}_k$ actually observes the losses; note, $N_F = |F|$ is equal to $T$ in full feedback and to the number queries issued in the partial information model.

**Lemma 3.1.** *For any determinstic algorithm $\mathcal{A}_k$ which issues at most $k$ best-action queries, we have:*

$$R_T^+(\mathcal{A}_k) + R_T^-(\mathcal{A}_k) \geq \exp\left(-\frac{5\varepsilon^2}{q}\mathbb{E}^+\left[N_F\right]\right) \cdot \frac{(T-k)\varepsilon}{2} - 2(q-\varepsilon)k.$$

*Proof.* The best action $i^*$ under $\mathbb{P}^+$, respectively $\mathbb{P}^-$ is the first, respectively the second, one, with an expected loss of

$$\mathbb{E}^\pm\left[\ell_t(i^*)\right] = \tfrac{1}{2} + q - \varepsilon. \tag{4}$$

On the other hand, the best realized action $i_t^*$ yields an expected loss of

$$\mathbb{E}^\pm\left[\ell_t(i_t^*)\right] = \mathbb{E}^\pm\left[\min\{\ell_t(1), \ell_t(2)\}\right] = -(q-\varepsilon). \tag{5}$$

Moreover, if the algorithm chooses a suboptimal action $i_t \neq i^*$, its expected instantaneous regret is:

$$\mathbb{E}^+\left[\ell_t(2)\right] - \mathbb{E}^+\left[\ell_t(i^*)\right] = \mathbb{E}^-\left[\ell_t(1)\right] - \mathbb{E}^-\left[\ell_t(i^*)\right] = 2\varepsilon. \tag{6}$$

Let now $N_+$, respectively $N_-$, be the random variable that counts the number of times that $\mathcal{A}_k$ selects action 1, respectively 2, in times that are not in $F$ (i.e., where the choice of 1 is not due to a query). Combining (4),(5), and (6), we have the following:

$$R_T^\pm(\mathcal{A}_k) \geq \mathbb{E}^\pm\left[\sum_{t \notin F}(\ell_t(i_t) - \ell_t(i^*)) + \sum_{t \in F}(\ell_t(i_t^*) - \ell_t(i^*))\right]$$
$$= 2\varepsilon \cdot \mathbb{E}^\pm\left[N_\mp\right] - (q-\varepsilon)\mathbb{E}^\pm\left[N_F\right].$$

Since $N_F \leq k$, and $N_+ + N_- \leq T - k$, we have the following bound on the regret:

$$R_T^+(\mathcal{A}_k) \geq \mathbb{P}^+\left[N_+ \leq \frac{T-k}{2}\right] \cdot (T-k)\varepsilon - (q-\varepsilon)k$$

$$R_T^-(\mathcal{A}_k) \geq \mathbb{P}^-\left[N_+ > \frac{T-k}{2}\right] \cdot (T-k)\varepsilon - (q-\varepsilon)k.$$

Summing the above two expressions, we obtain

$$R_T^+(\mathcal{A}_k) + R_T^-(\mathcal{A}_k) \geq \left(\mathbb{P}^+\left[N_+ \leq \frac{T-k}{2}\right] + \mathbb{P}^-\left[N_+ > \frac{T-k}{2}\right]\right) \cdot (T-k)\varepsilon - 2(q-\varepsilon)k. \tag{7}$$

At this point, we apply the Bretagnolle-Huber Inequality [Theorem 14.2 in Lattimore and Szepesvári, 2020] to bound the first term on the right-hand side of the above inequality:

$$\mathbb{P}^+\left[N_+ \le \frac{T-k}{2}\right] + \mathbb{P}^-\left[N_+ > \frac{T-k}{2}\right] \ge \frac{1}{2} \cdot \exp\left(-\mathcal{D}_{\mathrm{KL}}\left(\mathbb{P}^+_{Z_{\le T}}, \mathbb{P}^-_{Z_{\le T}}\right)\right),$$

where $\mathbb{P}^\pm_{Z_{\le T}}$ is the push-forward measure on all the possible sequences of feedback observed by $\mathcal{A}_k$ under $\mathbb{P}^\pm$. The lemma is concluded by combining the above inequality with the following claim, and plugging it into (7).

**Claim 3.2.** *It holds that* $\mathcal{D}_{\mathrm{KL}}\left(\mathbb{P}^+_{Z_{\le T}}, \mathbb{P}^-_{Z_{\le T}}\right) \le \frac{5\varepsilon^2}{q} \mathbb{E}^+\left[N_F\right].$

*Proof.* Let us observe that, once we fix the feedback history until time $t-1$, i.e., fix a realization of the feedback $Z_{\le t-1}$, we have that

$$\mathcal{D}_{\mathrm{KL}}\left(\mathbb{P}^+_{Z_t|Z_{\le t-1}}, \mathbb{P}^-_{Z_t|Z_{\le t-1}}\right) = \mathbb{I}\{t \in F\} \cdot \mathcal{D}_{\mathrm{KL}}\left(\mathbb{P}^+_{Z_t|t\in F}, \mathbb{P}^-_{Z_t|t\in F}\right),$$

where $\mathbb{P}^+_{Z_t|Z_{\le t-1}}$ (respectively $\mathbb{P}^-_{Z_t|Z_{\le t-1}}$) is the push-forward measure over $\{0,1\}^2$ when losses are drawn according to $\mathbb{P}^+$ (respectively $\mathbb{P}^-$), conditioning on the previous observations. The equality above holds because (i) algorithm $\mathcal{A}_k$ observes feedback if and only $t \in F$ (by definition), and (ii) $\mathcal{A}_k$ is deterministic and whether or not $t \in F$ may only depend on the past. We now upper bound the KL-divergence term above:

$$\mathcal{D}_{\mathrm{KL}}\left(\mathbb{P}^+_{Z_t|Z_{\le t-1}}, \mathbb{P}^-_{Z_t|Z_{\le t-1}}\right) = (q+\varepsilon) \cdot \log\left(1 + \frac{2\varepsilon}{q-\varepsilon}\right) + (q-\varepsilon) \cdot \log\left(1 - \frac{2\varepsilon}{q+\varepsilon}\right)$$

$$\le (q+\varepsilon) \cdot \frac{2\varepsilon}{q-\varepsilon} - (q-\varepsilon) \cdot \frac{2\varepsilon}{q+\varepsilon} = \frac{4\varepsilon^2 q}{q^2-\varepsilon^2} \le \frac{5\varepsilon^2}{q},$$

where the first inequality follows from $\log(1+z) \le z$ for all $z \in \mathbb{R}$, and the last holds as long as we choose $\varepsilon < \frac{q}{\sqrt{5}}$. To complete our derivation, we express the overall KL divergence exploiting the tower property of conditional expectation:

$$\mathcal{D}_{\mathrm{KL}}\left(\mathbb{P}^+_{Z_{\le T}}, \mathbb{P}^-_{Z_{\le T}}\right) = \sum_{t\in[T]} \mathbb{E}^+\left[\mathcal{D}_{\mathrm{KL}}\left(\mathbb{P}^+_{Z_t|Z_{\le t-1}}, \mathbb{P}^-_{Z_t|Z_{\le t-1}}\right)\right] \le \frac{5\varepsilon^2}{q} \mathbb{E}^+\left[N_F\right],$$

where expectation is taken over all possible feedback realizations $Z_{\le t-1}$, and the last inequality follows by earlier derivations. $\square$

This concludes the proof of the lemma. $\square$

We now show how to use the above lemma to derive the lower bounds. We start with full feedback.

**Theorem 3.3.** *In the full feedback query model, for all $k \in [T]$, we have the following lower bounds:*

- *For any algorithm $\mathcal{A}_k$ that has access to $k < c_0\sqrt{T}$ queries, it holds that $R_T(\mathcal{A}_k) \ge c_0 \frac{\sqrt{T}}{4}$.*

- *For any algorithm $\mathcal{A}_k$ that has access to $k \ge c_0\sqrt{T}$ queries, it holds that $R_T(\mathcal{A}_k) \ge c_1 \frac{T}{k}$.*

*Where $c_0 = 1/(e^8\sqrt{5})$ and $c_1 = 1/(320e^2)$ are universal constants.*

*Proof.* In full feedback, the algorithm always observes the losses, so that the feedback variable $Z_t = \ell_t$ and $N_F = T$. We prove the Theorem via Yao's minimax Theorem: we prove that any deterministic algorithm $\mathcal{A}_k$ fails against the random instance composed as follows: with probability $1/2$ the losses are drawn i.i.d. according to $\mathbb{P}^+$, otherwise, they are drawn i.i.d. according to $\mathbb{P}^-$. We can then apply Lemma 3.1 and obtain that the expected regret of $\mathcal{A}_k$ against such mixture is equal to

$$\mathbb{E}\left[R_T(\mathcal{A}_k)\right] = \frac{1}{2}(R_T^+(\mathcal{A}_k) + R_T^-(\mathcal{A}_k)) \ge \exp\left(-\frac{5\varepsilon^2}{q}T\right) \cdot \frac{(T-k)\varepsilon}{2} - 2(q-\varepsilon)k. \quad (8)$$

Now, if $k < c_0\sqrt{T}$, we set $\varepsilon = \frac{2}{\sqrt{5T}}$ and $q = \frac{1}{4}$ in (8) to get

$$\mathbb{E}\left[R_T(\mathcal{A}_k)\right] = \frac{1}{2}(R_T^+(\mathcal{A}_k) + R_T^-(\mathcal{A}_k)) \geq \frac{\sqrt{T}}{2e^8\sqrt{5}} - \frac{\sqrt{T}}{4e^8\sqrt{5}} \geq \frac{\sqrt{T}}{4e^8\sqrt{5}},$$

Otherwise, if $k \geq c_0\sqrt{T}$, then consider the following choice of the parameters: $\varepsilon = \frac{1}{40ek} + \frac{4e-1}{40eT}$ and $q = 5\varepsilon^2 T = \frac{T}{320e^2k^2} + \frac{4e-1}{160e^2k} + \frac{(4e-1)^2}{320e^2T}$. Plugging these parameters in (8), we get:

$$\mathbb{E}\left[R_T(\mathcal{A}_k)\right] = \frac{1}{2}(R_T^+(\mathcal{A}_k) + R_T^-(\mathcal{A}_k)) \geq \frac{T\varepsilon}{4e} - 5\varepsilon^2 kT + \left(1 - \frac{1}{4e}\right) \cdot \varepsilon k$$

$$= \frac{T}{320e^2k} + \frac{4e-1}{160e^2} + \frac{(4e-2)^2k}{320e^2T} \geq \frac{T}{320e^2k}. \qquad \square$$

A similar analysis can be carried over for the label efficient setting.

**Theorem 3.4.** *In the label efficient feedback model, for all $k \in [T]$, we have the following lower bounds:*

- *For any algorithm $\mathcal{A}_k$ that has access to $k < c_0\frac{T}{\sqrt{k}}$ queries, it holds that $R_T(\mathcal{A}_k) \geq c_0\frac{T}{4\sqrt{k}}$.*

- *For any algorithm $\mathcal{A}_k$ that has access to $k \geq c_0\frac{T}{\sqrt{k}}$ queries, it holds that $R_T(\mathcal{A}_k) \geq c_1\frac{T^2}{k^2}$.*

*Where $c_0 = {}^1/{(e^8\sqrt{5})}$ and $c_1 = {}^1/{(320e^2)}$ are universal constants.*

*Proof.* In the label efficient feedback model, $Z_t$ is meaningful only for times in $F$. We then prove the result by Yao's minimax principle, by showing that any deterministic algorithm $\mathcal{A}_k$ which issues at most $k$ queries suffers the desired regret against the instance that uniformly chooses between $\mathbb{P}^+$ and $\mathbb{P}^-$. We can apply Lemma 3.1 (nothing that $N_F \leq k$) to get:

$$\mathbb{E}\left[R_T(\mathcal{A}_k)\right] = \frac{1}{2}(R_T^+(\mathcal{A}_k) + R_T^-(\mathcal{A}_k)) \geq \exp\left(-\frac{5\varepsilon^2}{q}k\right) \cdot \frac{(T-k)\varepsilon}{4} - (q - \varepsilon)k.$$

Once again, we have two cases. If $k < c_0\frac{T}{\sqrt{k}}$, we choose $\varepsilon = \frac{2}{\sqrt{5k}}$ and $q = \frac{1}{4}$ to get

$$\mathbb{E}\left[R_T(\mathcal{A}_k)\right] = \frac{1}{2}(R_T^+(\mathcal{A}_k) + R_T^-(\mathcal{A}_k)) \geq \frac{T}{2e^8\sqrt{5k}} - \frac{T}{4e^8\sqrt{5k}} \geq \frac{T}{4e^8\sqrt{5k}}.$$

Otherwise, if $k \geq c_0\frac{T}{\sqrt{k}}$, and we choose $\varepsilon = \frac{T}{40ek^2} + \frac{4e-1}{40ek}$ and $q = 5\varepsilon^2 k = \frac{T^2}{320e^2k^3} + \frac{(4e-1)T}{160e^2k^2} + \frac{(4e-1)^2}{320e^2k}$, to obtain

$$\mathbb{E}\left[R_T(\mathcal{A}_k)\right] = \frac{1}{2}(R_T^+(\mathcal{A}_k) + R_T^-(\mathcal{A}_k)) \geq \frac{T\varepsilon}{4e} - 5\varepsilon^2 k^2 + \left(1 - \frac{1}{4e}\right) \cdot \varepsilon k$$

$$= \frac{T^2}{320e^2k^2} + \frac{T(4e-1)}{160e^2k} + \frac{(4e-1)^2}{320e^2} \geq \frac{T^2}{320e^2k^2}. \quad \square$$

## 4 Conclusions

Our work introduces *best-action queries* in the context of online learning. We provide tight minimax regret in both the full feedback model and in the label efficient one. We establish that leveraging a sublinear number of best action queries is enough to improve significantly the regret rates achievable *without* best-action queries. Promising avenues for future research involve integrating best-action queries with diverse feedback forms, extending beyond full feedback, such as bandit feedback, partial monitoring, and feedback graphs (where, in particular, Lemma 2.1 does not hold). Moreover, our work only studies the case where queries are *perfect*, i.e., the queried oracle gives the correct identity of the best action at that time step with probability $1$. Imagining a *noisy* oracle that gives the correct identity of the best action only with probability ${}^1/n + \delta$ is also an interesting future direction this work leaves open.

## Acknowledgments

Andrea Celli is partially supported by MUR - PRIN 2022 project 2022R45NBB funded by the NextGenerationEU program. Federico Fusco, Stefano Leonardi and Matteo Russo are partially supported by the ERC Advanced Grant 788893 AMDROMA "Algorithmic and Mechanism Design Research in Online Markets", by the FAIR (Future Artificial Intelligence Research) project PE0000013, funded by the NextGenerationEU program within the PNRR-PE-AI scheme (M4C2, investment 1.3, line on Artificial Intelligence), and by the PNRR MUR project IR0000013-SoBigData.it. Stefano Leonardi and Matteo Russo are also partially supported by the MUR PRIN grant 2022EKNE5K (Learning in Markets and Society).

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

# A    Omitted Proofs

## A.1    Proof of Lemma 2.1

**Lemma 2.1.** *Consider the Hedge algorithm* $\text{Hedge}_\eta(\tilde{\ell})$ *run on loss sequence* $\tilde{\ell} \in [0, U]^{n \times T}$ *with learning rate* $\eta < 1/U$. *Then, for all action* $i \in [n]$, *it holds that,*

$$\tilde{L}_T(\text{Hedge}_\eta(\tilde{\ell})) \leq \frac{1}{1 - U\eta} \cdot \left( \tilde{L}_T(i) + \frac{\log n}{\eta} \right),$$

*where* $\tilde{L}_T(\text{Hedge}_\eta(\tilde{\ell})) = \sum_{t \in [T], i \in [n]} p_t(i) \cdot \tilde{\ell}_t(i)$ *is the expected cumulative loss of* $\text{Hedge}_\eta(\tilde{\ell})$ *and* $\tilde{L}_T(i)$ *is the cumulative loss of action* $i$.

*Proof.* We have that, by definition,

$$W_{T+1} \geq w_{T+1}(i) w_t(i) \cdot \exp\left( -\eta \tilde{\ell}_t(i) \right) = \prod_{t \in [T]} \exp\left( -\eta \tilde{\ell}_t(i) \right) = \exp\left( -\eta \tilde{L}_T(i) \right).$$

We also know that

$$W_{t+1} \leq W_t \cdot \left( 1 - \eta \sum_{i \in [n]} p_t(i) \cdot \tilde{\ell}_t(i) + \eta^2 \sum_{i \in [n]} p_t(i) \cdot \tilde{\ell}_t^2(i) \right).$$

Thus,

$$W_{T+1} \leq n \cdot \prod_{t \in [T]} \left( 1 - \eta \sum_{i \in [n]} p_t(i) \cdot \tilde{\ell}_t(i) + \eta^2 \sum_{i \in [n]} p_t(i) \cdot \tilde{\ell}_t^2(i) \right),$$

which combined with the earlier bound and taking logarithms of both sides, gives

$$-\eta \tilde{L}_T(i) \leq \log n + \sum_{t \in [T]} \log \left( 1 - \eta \sum_{i \in [n]} p_t(i) \cdot \tilde{\ell}_t(i) + \eta^2 \sum_{i \in [n]} p_t(i) \cdot \tilde{\ell}_t^2(i) \right)$$

$$\leq \log n - \eta \sum_{t \in [T], i \in [n]} p_t(i) \cdot \tilde{\ell}_t(i) + \eta^2 \sum_{t \in [T], i \in [n]} p_t(i) \cdot \tilde{\ell}_t^2(i)$$

$$\leq \log n - (\eta - U\eta^2) \sum_{t \in [T], i \in [n]} p_t(i) \cdot \tilde{\ell}_t(i).$$

The second inequality above follows from $\log(1 + z) \leq z$ for $z \in \mathbb{R}$, and the third by observing that $\tilde{\ell}_t^2(i) \leq U \tilde{\ell}_t(i)$, since $\tilde{\ell}_t(i) \in [0, U]$. The lemma follows by rearranging the terms above. $\qquad\square$

# B    Best-Action Queries in the Stochastic i.i.d. Setting

In this section, we show how the results provided in Section 2 can be obtained in the stochastic i.i.d. setting (defined next) using the Follow-The-Leader algorithm, albeit a suboptimal dependence in $n$.

In the stochastic i.i.d. setting, each action is associated with a fixed but unknown distribution $D_i$ supported in $[0, 1]$. Action $i$'s loss at time step $t$, $\ell_t(i)$, is drawn independently from distribution $D_i$. We denote with $\mu(i)$ the expected loss of action $i$, and with $i^* = \arg\min_{i \in [n]} \mu(i)$ be the action of lowest expected loss. We measure the performance of a learning algorithm $\mathcal{A}$, by considering its regret with respect to the best action distribution:

$$R_T(\mathcal{A}) = \sum_{t \in [T]} \mathbb{E}\left[ \ell_t(i_t) - \ell_t(i^*) \right].$$

We denote by $\Delta_i = \mu(i) - \mu(i^*)$ be the gap between the expected loss of the best action and that of action $i$, and by $\Psi_i = \mathbb{E}\left[ |\ell(i) - \ell(i^*)| \right]$ the expected absolute value of such gap (note, we omit the dependence on time as the losses are drawn i.i.d. across time). Whenever the learner issues a query, then the identity of the action $i_t^*$ with best *realized* loss is revealed, so that the learner gets, in expectation, $\mathbb{E}\left[ \ell_t(i_t^*) \right] = \mathbb{E}\left[ \min_{i \in [n]} \ell_t(i) \right]$.

## B.1 Useful Facts and Observations

The following facts and simple results are particularly useful for analyzing algorithms in the stochastic generation model.

**Fact B.1** (Bernstein's Inequality). *Let $Y_1, \ldots, Y_m$ be independent random variables such that $\sum_{\tau=1}^{m} \mathbb{E}[Y_\tau] = \mu$ and $\mathbb{P}[|Y_\tau| \leq c] = 1$, for $c > 0$ and all $\tau \in [m]$. Then, for $\gamma > 0$, we have*

$$\mathbb{P}\left[\sum_{\tau=1}^{m} Y_\tau - \mu \geq \gamma\right] \leq \exp\left(-\frac{3\gamma^2}{6\sigma^2 + 2c\gamma}\right),$$

*where $\sigma^2 = \sum_{\tau=1}^{m} \mathbb{E}[Y_\tau^2] - \mathbb{E}[Y_\tau]^2$. Moreover, when $Y_\tau$'s are also identically distributed, we have $\sigma^2 = m(\mathbb{E}[Y_\tau^2] - \mathbb{E}[Y_\tau]^2) \leq m\mathbb{E}[|Y_\tau|] =: m\Psi$. Thus,*

$$\mathbb{P}\left[\sum_{\tau=1}^{m} Y_\tau - \mu \geq \gamma\right] \leq \exp\left(-\frac{3\gamma^2}{6m\Psi + 2c\gamma}\right).$$

**Proposition B.2.** *Let $T, n \in \mathbb{N}$ be the time horizon and number of experts respectively, with $T \geq n-1$. For all distributions $D_i$ of actions losses, and any algorithm $\mathcal{A}$, its regret is*

$$R_T(\mathcal{A}_k) = \sum_{t=k+1}^{T} \Delta_{i_t} - \frac{k}{2} \cdot (\Psi - \Delta), \tag{9}$$

*where $\Psi := \mathbb{E}[|\min_{i \neq i^*} \ell(i) - \ell(i^*)|]$ and $\Delta := \mathbb{E}[\min_{i \neq i^*} \ell(i) - \ell(i^*)]$.*

*Proof.* We first observe that, by the i.i.d. assumption, we can always make the algorithm query in the first $k$ time steps, without losing generality. Therefore, as $Q = \{t \mid t \leq k\}$, the algorithm suffers $\mathbb{E}[\min_{i \in [n]} \ell_t(i)]$ in the first $k$ time steps, which, by independence, gives

$$\sum_{t \leq k} \mathbb{E}\left[\min_{i \in [n]} \ell_t(i)\right] = k\mathbb{E}\left[\min_{i \in [n]} \ell(i)\right].$$

We also have the following useful observation that allows us to rewrite the maximum in a convenient form. Namely,

$$\min_{i \in [n]} \ell(i) = \frac{\ell(i^*) + \min_{i \neq i^*} \ell(i) - \left|\min_{i \neq i^*} \ell(i) - \ell(i^*)\right|}{2}. \tag{10}$$

The claim follows by the definitions of $\Psi$ and $\Delta$. $\qquad\square$

Thus, in the stochastic case, the goal of a regret-minimizing algorithm is to minimize the first summand above since the second is characteristic of the instance at hand.

**Observation B.3.** *Let $T, n \in \mathbb{N}$ be the time horizon and number of experts respectively, with $T \geq n - 1$. For all distributions $D_i$ of actions losses, $\Psi_i - \Delta_i \leq \Psi - \Delta$.*

*Proof.* Since $\mathbb{E}[\min_{j \in [n]} \ell(j)] \leq \mathbb{E}[\min\{\ell(i^*), \ell(i)\}]$ for all $i \in [n]$, we have

$$\Delta - \Psi = 2 \cdot \mathbb{E}\left[\min_{j \in [n]} \ell(j) - \ell(i^*)\right] \leq 2 \cdot \mathbb{E}[\min\{\ell(i), \ell(i^*)\} - \ell(i^*)] = \Delta_i - \Psi_i,$$

where the last equality also follows from Equation (10) applied to actions $i, i^*$ only. $\qquad\square$

## B.2 Label Efficient Feedback:Explore-Then-Commit with $\Omega(T^{2/3})$ Best-Action Queries

We consider the label efficient feedback case first, where feedback is given only during querying time steps. The techniques used in this section are key for the full feedback case, and easier to illustrate.

We provide a simple variation of the Explore-Then-Commit (ETC) algorithm Perchet et al. [2015] that, in the first $k \geq \Omega(T^{2/3})$ time steps, is given free access to the identity of the best action before committing to the action to select. We have the following theorem:

---

**Algorithm 2** Explore-Then-Commit with Best-Action Queries

---

**Input:** Sequence of losses $\ell_t(i) \sim D_i$ and $k \in [T]$ query budget
**for** $t \in [T]$ **do**
    **if** $t \leq k$ **then**
        Observe $i_t^* = \arg\min_{i \in [n]} \ell_t(i)$
        Select action $i_t^*$
        Observe $\ell_t(i)$ for all $i \in [n]$
    **else**
        Select action $j = \arg\min_{i \in [n]} \bar{\mu}_k(i)$

---

**Theorem B.4.** *Let $T, n \in \mathbb{N}$ be the time horizon and number of actions respectively, with $T \geq n - 1$. For all distributions $D_i$ of actions losses, Algorithm 2 with query access $k \in [4T/9]$ guarantees*

$$R_T(\mathcal{A}_k) \leq \min\left\{ 3T\sqrt{\frac{\log 2nT}{2k}}, \frac{2nT^2 \ln T}{k^2} \right\}.$$

We split the proof of this theorem in two lemmas that immediately imply it, one for $k < T^{2/3}$ and one otherwise.

**Lemma B.5.** *If $k < T^{2/3}$, then $R_T(\mathcal{A}_k) \leq 3T\sqrt{\frac{\log 2nT}{2k}}$.*

*Proof.* For all actions $i$, we define the clean event as

$$\mathcal{E}_i = \{ |\bar{\mu}_k(i) - \mu(i)| < \varepsilon \},$$

where $\varepsilon = \sqrt{\frac{\log 2nT}{2k}}$. Similarly, let $\mathcal{E}$ be the intersection of all the $\mathcal{E}_i$'s for $i \in [n]$. By Hoeffding's inequality [Equation (5.8) in Chapter 5 of Lattimore and Szepesvári, 2020], we have the following:

$$\mathbb{P}\left[\bar{\mathcal{E}}\right] = \mathbb{P}\left[\cup_{i \in [n]} \bar{\mathcal{E}}_i\right] \leq \sum_{i \in [n]} \mathbb{P}\left[\bar{\mathcal{E}}_i\right] \leq 2n \cdot \exp\left(-2k\varepsilon^2\right) \leq \tfrac{1}{T}.$$

Consider now the expected instantaneous regret suffered by ETC at time $t + 1$. We have two cases: Either the clean event holds, so that the instantaneous regret is at most $2\varepsilon_t$, or it does not, in which case the instantaneous regret is at most 1. By the law of total probability, we have the following:

$$\mathbb{E}\left[\ell_{t+1}(i_t) - \ell_{t+1}(i)\right] \leq \mathbb{P}\left[\bar{\mathcal{E}}_t\right] + \mathbb{E}\left[\ell_{t+1}(i_t) - \ell_{t+1}(i)|\mathcal{E}_t\right] \leq \tfrac{1}{T} + 2\varepsilon_t \leq 3\varepsilon_t.$$

Summing the expected instantaneous regrets for all $t$ yields the desired bound of

$$R_T(\mathcal{A}_k) \leq 3T\sqrt{\frac{\log 2nT}{2k}},$$

as we use the lower bound $\Psi - \Delta \geq 0$ to say that in the first $k$ times the regret the algorithm suffers is at most 0. □

**Lemma B.6.** *If $T^{2/3} \leq k \leq 4T/9$, then $R_T(\mathcal{A}_k) \leq \frac{2nT^2 \ln T}{k^2}$.*

*Proof.* We start by showing the statement for 2 actions, and then generalize to $n$.

**The case of 2 actions.** Let us recall that, by Proposition B.2, we have

$$R_T(\mathcal{A}_k) = |\{t \geq k + 1 \mid i_t = 2\}| \cdot \Delta - \frac{k}{2} \cdot (\Psi - \Delta),$$

where we have assumed that $\mu_1 \leq \mu_2$, without loss of generality. We suffer positive regret in the last $T - k$ time steps when we select action 2 over 1, which happens if and only if $\bar{\mu}_k(1) \leq \bar{\mu}_k(2)$. Hence, the first summand of the above regret expression simply becomes $\Delta(T - k) \cdot \mathbb{P}\left[\bar{\mu}_k(2) \leq \bar{\mu}_k(1)\right]$.

Note that

$$\mathbb{P}\left[\bar{\mu}_k(2) \leq \bar{\mu}_k(1)\right] = \mathbb{P}\left[\sum_{t=1}^{k} \ell_t(1) - \ell_t(2) \geq 0\right],$$

so if we define $g_t = \ell_t(1) - \ell_t(2)$, whose expected value is $\mathbb{E}[g_t] = -\Delta$ and $\mathbb{P}[|g_t| \leq 1] = 1$, then, by Fact B.8,

$$\mathbb{P}\left[\sum_{t=1}^{k} g_t \geq 0\right] = \mathbb{P}\left[\sum_{t=1}^{k} g_t + k\Delta \geq k\Delta\right]$$

$$\leq \exp\left(-\frac{3k^2\Delta^2}{6k\Psi + 2k\Delta}\right)$$

$$\leq \exp\left(-\frac{k\Delta^2}{2\Psi + \Delta}\right).$$

We distinguish two cases, namely $\frac{\Delta^2}{2\Psi+\Delta} \leq \frac{\ln T}{k}$ and $\frac{\Delta^2}{2\Psi+\Delta} > \frac{\ln T}{k}$. In the latter case, we get that

$$R_T(\mathcal{A}_k) = \Delta(T-k) \cdot \mathbb{P}[\bar{\mu}_k(2) \leq \bar{\mu}_k(1)] - \frac{k}{2} \cdot (\Psi - \Delta)$$

$$\leq \Delta T \cdot \exp\left(-\frac{k\Delta^2}{2\Psi + \Delta}\right) - \frac{k}{2} \cdot (\Psi - \Delta)$$

$$\leq \Delta T \cdot \exp\left(-\ln T\right) = \Delta \leq 1.$$

Otherwise, we obtain that $\Psi \geq \frac{\Delta}{2} \cdot \left(\frac{k\Delta}{\ln T} - 1\right)$, and so

$$R_T(\mathcal{A}_k) = \Delta(T-k) \cdot \mathbb{P}[\bar{\mu}_k(2) \leq \bar{\mu}_k(1)] - \frac{k}{2} \cdot (\Psi - \Delta)$$

$$\leq \Delta T - \frac{k\Delta}{4} \cdot \left(\frac{k\Delta}{\ln T} - 3\right)$$

$$\leq \left(\frac{T^2}{k^2} + \frac{3T}{4k} - \frac{27}{16}\right) \cdot \ln T$$

$$\leq \frac{2T^2 \ln T}{k^2}.$$

The first inequality holds since the maximizing $\Delta = \frac{4T-3k}{2k^2} \cdot \ln T$, which is consistent with $0 \leq \Delta \leq \Psi \leq 1$ since $T^{2/3} \leq k \leq 4T/9$.

**Generalizing to $n$ actions.** We again assume that the first is the best action, and apply Proposition B.2, to obtain

$$R_T(\mathcal{A}_k) = \sum_{\substack{t=k+1 \\ j \neq 1}}^{T} \mathbb{E}\left[\mathbb{I}\left\{\bar{\mu}_k(j) \geq \max_{i \neq j} \bar{\mu}_k(i)\right\}(\ell_t(j) - \ell_t(1))\right] - \frac{k}{2} \cdot (\Psi - \Delta)$$

$$= (T-k) \cdot \sum_{j \neq 1} \Delta_j \cdot \mathbb{P}\left[\bar{\mu}_k(j) \geq \max_{i \neq j} \bar{\mu}_k(i)\right] - \frac{k}{2} \cdot (\Psi - \Delta)$$

$$\leq T \cdot \sum_{j \neq 1} \Delta_j \cdot \mathbb{P}[\bar{\mu}_k(j) \geq \bar{\mu}_k(1)] - \frac{k}{2} \cdot (\Psi - \Delta),$$

which holds by the independence of realizations across time steps. Let us define $g_t(j) = \ell_t(1) - \ell_t(j)$, so that, $\Psi_j = \mathbb{E}[|g_t(j)|]$, $\mathbb{E}[g_t(j)] = -\Delta_j$ and $\mathbb{P}[|g_t(j)| \leq 1] = 1$. Then, by Fact B.8, we obtain

$$\mathbb{P}[\bar{\mu}_k(j) \geq \bar{\mu}_k(1)] = \mathbb{P}\left[\sum_{t=1}^{k} g_t(j) \geq 0\right] \leq \exp\left(-\frac{k\Delta_j^2}{2\Psi_j + \Delta_j}\right).$$

Recall that, by Observation B.3, $\Psi - \Delta \geq \Psi_j - \Delta_j$. Therefore, let us sum and subtract to the regret upper bound the sum $\frac{k}{2n} \cdot \sum_{j \neq 1}(\Psi_j - \Delta_j)$, and get

$$R_T(\mathcal{A}_k) \leq \sum_{j \neq 1}\left(T\Delta_j \cdot \exp\left(-\frac{k\Delta_j^2}{2\Psi_j + \Delta_j}\right) - \frac{k}{2n} \cdot (\Psi_j - \Delta_j)\right)$$

$$+ \frac{k}{2} \cdot \left( \frac{1}{n} \cdot \sum_{j \neq 1} (\Psi_j - \Delta_j) - (\Psi - \Delta) \right)$$

$$\leq \sum_{j \neq 1} \left( T\Delta_j \cdot \exp\left( -\frac{k\Delta_j^2}{2\Psi_j + \Delta_j} \right) - \frac{k}{2n} \cdot (\Psi_j - \Delta_j) \right).$$

For each $j \neq 1$, an identical derivation to the one in the case of two actions would yield the same regret rate, with $k/n$ in place of $k$. Thus, overall, we obtain

$$R_T(\mathcal{A}_k) \leq \frac{2nT^2 \ln T}{k^2},$$

which concludes the proof. $\qquad\square$

## B.3 Full Feedback: Follow-The-Leader with $\Omega\left(\sqrt{T}\right)$ Best-Action Queries

We provide an algorithm that, equipped with best-action queries for $k$ time steps and with full feedback, achieves regret $O(T/k)$ for $k \geq \Omega(\sqrt{T})$, and $O(\sqrt{T})$ otherwise. This means that, with the addition of feedback, fewer queries are needed to switch to a much smaller regret rate.

The idea is to use the Follow-The-Leader paradigm Auer et al. [2003]. That is, we select the best action in the first $k$ time steps since queries give us its identity for free, and successively, we select the action maximizing the empirical average so far, $\bar{\mu}_{t-1}(i) = \frac{1}{t-1} \cdot \sum_{\tau=1}^{t-1} \ell_\tau(i)$.

---

**Algorithm 3** Follow-The-Leader with Best-Action Queries

**Input:** Sequence of losses $\ell_t(i) \sim D_i$ and $k \in [T]$ query budget
**for** $t \in [T]$ **do**
    **if** $t \leq k$ **then**
        Observe $i_t^* = \arg\min_{i \in [n]} \ell_t(i)$
        Select action $i_t^*$
        Observe $\ell_t(i)$ for all $i \in [n]$
    **else**
        Select action $j = \arg\min_{i \in [n]} \bar{\mu}_{t-1}(i)$

---

We have the following guarantee:

**Theorem B.7.** *Let $T, n \in \mathbb{N}$ be the time horizon and number of actions respectively, with $T \geq n - 1$. For all distributions $D_i$ of actions losses, Algorithm 3 with query access $k \geq 2\sqrt{T}$ guarantees*

$$R_T(\mathcal{A}_k) \leq \min\left\{ 3\sqrt{2T \log 2nT}, \frac{5nT}{k} \right\}.$$

For our purposes, only a weaker version of Fact B.1 is needed:

**Fact B.8.** *Let $Y_1, \ldots, Y_m$ be i.i.d. random variables such that $\mathbb{E}[Y_\tau] = -\Delta_Y$, $\mathbb{E}[|Y_\tau|] = \Psi_Y$, and $\mathbb{P}[|Y_\tau| \leq 1] = 1$, for all $\tau \in [m]$. Then, it holds that*

$$\mathbb{P}\left[ \sum_{\tau=1}^m Y_\tau \geq 0 \right] \leq \exp\left( -\frac{m\Delta_Y^2}{2\Psi_Y + \Delta_Y} \right).$$

We split the proof of this theorem in two lemmas that immediately imply it, one for $k < 2\sqrt{T}$ and one otherwise. Next, we subsume the notation of Theorem B.7.

**Lemma B.9.** *If $k < 2\sqrt{T}$, then $R_T(\mathcal{A}_k) \leq 3\sqrt{2T \log 2nT}$.*

*Proof.* For all times $t$ and actions $i$, we define the clean event as

$$\mathcal{E}_{i,t} = \{ |\bar{\mu}_t(i) - \mu(i)| < \varepsilon_t \},$$

where $\varepsilon_t = \sqrt{\frac{\log 2nT}{2t}}$. Similarly, let $\mathcal{E}_t$ be the intersection of all the $\mathcal{E}_{i,t}$'s for $i \in [n]$. By Hoeffding's inequality [Equation (5.8) in Chapter 5 of Lattimore and Szepesvári, 2020], we have the following:

$$\mathbb{P}\left[\bar{\mathcal{E}}_t\right] = \mathbb{P}\left[\cup_{i\in[n]}\bar{\mathcal{E}}_{i,t}\right] \leq \sum_{i\in[n]} \mathbb{P}\left[\bar{\mathcal{E}}_{i,t}\right] \leq 2n \cdot \exp\left(-2t\varepsilon_t^2\right) \leq \tfrac{1}{T}.$$

Consider now the expected instantaneous regret suffered by FTL at time $t+1$. We have two cases: Either the clean event holds, so that the instantaneous regret is at most $2\varepsilon_t$, or it does not, in which case the instantaneous regret is at most 1. By the law of total probability, we have the following:

$$\mathbb{E}\left[\ell_{t+1}(i_t) - \ell_{t+1}(i)\right] \leq \mathbb{P}\left[\bar{\mathcal{E}}_t\right] + \mathbb{E}\left[\ell_{t+1}(i_t) - \ell_{t+1}(i)|\mathcal{E}_t\right] \leq \tfrac{1}{T} + 2\varepsilon_t \leq 3\varepsilon_t.$$

Summing the expected instantaneous regrets for all $t$ yields the desired bound of

$$R_T(\mathcal{A}_k) \leq 3\sqrt{2T\log 2nT},$$

as we use the lower bound $\Psi - \Delta \geq 0$ to say that in the first $k$ times the regret the algorithm suffers is at most 0. $\qquad\square$

**Lemma B.10.** *If $k \geq 2\sqrt{T}$, then $R_T(\mathcal{A}_k) \leq \frac{5nT}{k}$.*

*Proof.* We start by showing the statement for 2 actions, and then generalize to $n$.

**The case of 2 actions.** We again assume that $\mu_1 \leq \mu_2$ without loss of generality. FTL's regret, thus, reads

$$
\begin{aligned}
R_T(\mathcal{A}_k) &= \mathbb{E}\left[\sum_{t=k+1}^{T} (\ell_t(2) - \ell_t(1)) \cdot \mathbb{I}\{\bar{\mu}_{t-1}(2) \leq \bar{\mu}_{t-1}(1)\}\right] - \frac{k}{2} \cdot (\Psi - \Delta) \\
&= \Delta \cdot \sum_{t=k+1}^{T} \mathbb{P}\left[\bar{\mu}_{t-1}(2) \leq \bar{\mu}_{t-1}(1)\right] - \frac{k}{2} \cdot (\Psi - \Delta) \\
&\leq \Delta \cdot \sum_{t=k+1}^{T} \exp\left(-\frac{t\Delta^2}{2\Psi + \Delta}\right) - \frac{k}{2} \cdot (\Psi - \Delta),
\end{aligned}
\tag{11}
$$

where the second equality follows from the independence of step $t$ from all the preceding steps, and the inequality by Fact B.8. We distinguish three cases, namely $\frac{\Delta^2}{2\Psi+\Delta} \leq \frac{1}{T}$, $\frac{1}{T} < \frac{\Delta^2}{2\Psi+\Delta} \leq \frac{\ln T}{k}$, and $\frac{\Delta^2}{2\Psi+\Delta} > \frac{\ln T}{k}$. In the first case, we have that $\Psi \geq \frac{\Delta}{2} \cdot (T\Delta - 1)$ and so,

$$
\begin{aligned}
R_T(\mathcal{A}_k) &\leq \Delta T - \frac{k\Delta}{4} \cdot (T\Delta - 3) \\
&\leq \frac{T}{k} + \frac{3}{2} + \frac{27k}{64T} \\
&\leq \frac{2T}{k},
\end{aligned}
$$

where the second inequality follows since the $\Delta$ maximizing the expression is $\Delta = \frac{2}{k} + \frac{3}{4T}$. This implies that $\Psi \geq \frac{T}{2k^2} - \frac{3}{32T} + \frac{1}{2k}$, which is consistent with $0 \leq \Delta \leq \Psi \leq 1$ since $k \geq 2\sqrt{T}$.

In the last case, $\frac{\Delta^2}{2\Psi+\Delta} > \frac{\ln T}{k}$, we get

$$
\begin{aligned}
R_T(\mathcal{A}_k) &= \Delta(T - k) \cdot \mathbb{P}\left[\bar{\mu}_{t-1}(2) \leq \bar{\mu}_{t-1}(1)\right] - \frac{k}{2} \cdot (\Psi - \Delta) \\
&\leq \Delta T \cdot \exp\left(-\frac{t\Delta^2}{2\Psi + \Delta}\right) - \frac{k}{2} \cdot (\Psi - \Delta) \\
&\leq \Delta T \cdot \exp\left(-\ln T\right) = \Delta \\
&\leq 1.
\end{aligned}
$$

We are left to show the case where $\frac{1}{T} < \frac{\Delta^2}{2\Psi+\Delta} \leq \frac{\ln T}{k}$. To this end, let us observe that, for all $T \geq 2$, it holds that

$$\sum_{t=m+1}^{T} \exp\left(-\frac{t}{r}\right) \leq r.$$

for all $1 \leq m, r \leq T$. This is the case because

$$\sum_{t=m+1}^{T} \exp\left(-\frac{t}{r}\right) = \frac{e^{-k/r} - e^{-T/r}}{e^{1/r} - 1} \leq re^{-k/r} \leq r,$$

since $1 + z \leq e^z$ for all $z \in \mathbb{R}$. Therefore,

$$R_T(\mathcal{A}_k) \leq 2\Delta \cdot \frac{2\Psi+\Delta}{\Delta^2} - \frac{k}{2}(\Psi - \Delta) = \Psi \cdot \left(\frac{4}{\Delta} - \frac{k}{2}\right) + 2 + \frac{k\Delta}{2}.$$

We distinguish two subcases: On the one hand, assume that $\Delta > \frac{8}{k}$, then $\frac{4}{\Delta} - \frac{k}{2} < 0$. Since $\Psi \geq \frac{k\Delta^2}{2\ln T} - \frac{\Delta}{2}$, we have

$$R_T(\mathcal{A}_k) \leq \left(\frac{k\Delta^2}{2\ln T} - \frac{\Delta}{2}\right) \cdot \left(\frac{4}{\Delta} - \frac{k}{2}\right) + 2 + \frac{k\Delta}{2}$$

$$= \frac{2k\Delta}{\ln T} - \frac{k\Delta^2}{4\ln T} + \frac{3k}{4}$$

$$\leq \frac{3\ln T}{8} + 4 + \frac{1}{\ln T}\left(8 - \frac{4}{k}\right) - \frac{1}{k} \cdot \left(\frac{\ln T}{16} - 1\right)$$

$$\leq \frac{\ln T}{2},$$

since the expression is maximized for $\Delta = \frac{8 + \ln T}{2k} > \frac{8}{k}$, as long as $T > e^8$. This implies that $\Psi \geq \frac{8}{k\ln T} - \frac{\ln T}{8k}$, which is consistent with $0 \leq \Delta \leq \Psi \leq 1$. On the other hand, $\Delta \leq \frac{8}{k}$, and thus $\frac{4}{\Delta} - \frac{k}{2} \geq 0$. We use that $\Psi \leq \frac{T\Delta^2 - \Delta}{2}$, and get

$$R_T(\mathcal{A}_k) \leq \frac{T\Delta^2 - \Delta}{2} \cdot \left(\frac{4}{\Delta} - \frac{k}{2}\right) + 2 + \frac{k\Delta}{2}$$

$$= 2T\Delta - \frac{kT\Delta^2}{4} + \frac{3k\Delta}{4}$$

$$\leq \frac{4T}{k} + \frac{9k}{16T}$$

$$\leq \frac{5T}{k},$$

since the expression is maximized for $\Delta = \frac{4}{k} + \frac{3}{2T} < \frac{8}{k}$. This implies that $\Psi \leq \frac{8T}{k^2} + \frac{4}{k} + \frac{3}{8T}$, which is consistent with $0 \leq \Delta \leq \Psi \leq 1$ since $k \geq 2\sqrt{T}$.

**Generalizing to $n$ actions.** The generalization is very similar to the one in the proof of B.4. Indeed, the regret equals

$$R_T(\mathcal{A}_k) = \sum_{\substack{t=k+1 \\ j \neq 1}}^{T} \mathbb{E}\left[\mathbb{I}\left\{\bar{\mu}_{t-1}(j) \leq \min_{i \neq j} \bar{\mu}_{t-1}(i)\right\}(\ell_t(j) - \ell_t(1))\right] - \frac{k}{2} \cdot (\Psi - \Delta)$$

$$= \sum_{t=k+1}^{T}\sum_{j\neq 1}\Delta_j \cdot \mathbb{P}\left[\bar{\mu}_{t-1}(j) \leq \min_{i \neq j} \bar{\mu}_{t-1}(i)\right] - \frac{k}{2} \cdot (\Psi - \Delta)$$

$$\leq \sum_{t=k+1}^{T}\sum_{j\neq 1}\Delta_j \cdot \mathbb{P}\left[\bar{\mu}_{t-1}(j) \leq \bar{\mu}_{t-1}(1)\right] - \frac{k}{2} \cdot (\Psi - \Delta)$$

$$\leq \sum_{j \neq 1} \Delta_j \cdot \sum_{t=k+1}^{T} \exp\left(-\frac{t\Delta_j^2}{2\Psi_j + \Delta_j}\right) - \frac{k}{2} \cdot (\Psi - \Delta),$$

where the last inequality follows by Fact B.8. Identically to the proof of Theorem B.4, we sum and subtract the term $\frac{k}{2n} \cdot \sum_{j \neq 1}(\Psi_j - \Delta_j)$, and get

$$
\begin{aligned}
R_T(\mathcal{A}_k) &\leq \sum_{j \neq 1} \left( \Delta_j \cdot \sum_{t=k+1}^{T} \exp\left(-\frac{t\Delta_j^2}{2\Psi_j + \Delta_j}\right) - \frac{k}{2n} \cdot (\Psi_j - \Delta_j) \right) \\
&\quad + \frac{k}{2} \cdot \left( \frac{1}{n} \cdot \sum_{j \neq 1}(\Psi_j - \Delta_j) - (\Psi - \Delta) \right) \\
&\leq \sum_{j \neq 1} \left( \Delta_j \cdot \sum_{t=k+1}^{T} \exp\left(-\frac{t\Delta_j^2}{2\Psi_j + \Delta_j}\right) - \frac{k}{2n} \cdot (\Psi_j - \Delta_j) \right).
\end{aligned}
$$

For each $j \neq 1$, an identical derivation to the one in the case of two actions would yield the same regret rate, with $k/n$ in place of $k$. Thus, overall, we obtain

$$R_T(\mathcal{A}_k) \leq \frac{5nT}{k},$$

which concludes the proof. □

## C  Further Related Work

**Correlated hints.**    A first model that is close to ours has been introduced by Dekel et al. [2017], which studies Online Linear Optimization when the learner has access to vectors correlated with the actual losses. Surprisingly, this type of hint guarantees an exponential improvement in the regret bound, which is logarithmic in the time horizon when the optimization domain is strongly convex. Subsequently, Bhaskara et al. [2020] generalize such results in the presence of *imperfect hints*, while Bhaskara et al. [2023b] prove that $\Omega(\sqrt{T})$ hints are enough to achieve the logarithmic regret bound. While these works share significant similarities with ours, we stress that their results are not applicable to the standard prediction with experts model, given the non-strong convexity of the probability simplex over the actions.

**Queries/Ordinal hints.**    Bhaskara et al. [2023b] studies online learning algorithms augmented with ordinal queries of the following type: a query takes in input a small subset of the actions and receives in output the identity of the best one. With this twist, it is possible to bring the regret down to $O(1)$ by observing only 2 experts losses in advance at each time. Although our best-action query is stronger (as it compares *all* the actions), our learner is constrained in the number of times it can issue such queries.

**Algorithms with predictions.**    Other works that explore the interplay between hints and feedback are [Bhaskara et al., 2021a, Shi et al., 2022, Cheng et al., 2023, Bhaskara and Munagala, 2023, Bhaskara et al., 2023a]. More broadly, our work follows the literature on enhancing performance through external information (predictions). This has been extensively applied to a variety of online problems to model partial information about the input sequence that can be fruitfully exploited if accurate for improving the performance of the algorithms. Examples include sorting [Bai and Coester, 2023], frequency estimation [Aamand et al., 2023], various online problems [Purohit et al., 2018, Gollapudi and Panigrahi, 2019] such as metrical task systems [Antoniadis et al., 2023b,c], graph coloring [Antoniadis et al., 2023a], caching [Lykouris and Vassilvitskii, 2021], scheduling [Lattanzi et al., 2020, Jiang et al., 2022], and several others.

