# OpenReview forum: "Online Learning with Sublinear Best-Action Queries"
_NeurIPS.cc/2024/Conference — NeurIPS 2024 poster_

### Official Review · Reviewer_tNck · 2024-07-08

**Soundness:** 3
**Presentation:** 2
**Contribution:** 3
**Rating:** 5
**Confidence:** 2

**Summary:**

In this paper, the authors studies the classical problem of online learning in the additional context of algorithms with predictions, or learning-augmented algorithms. In online learning, a learning algorithm must repeatedly select from a set of options, each associated with a loss function, and the goal is to minimize the regret, i.e., the difference between the (expected) loss of the algorithm and that of the best fixed option in hindsight. The loss function may be adversarially generated based on time and prior actions, which makes obtaining good regret traditionally hard. By utilizing predictions to give the learning algorithm some additional information about the unknown loss function, we may be able to obtain better regrets and break impossibility bounds.

The authors incorporate 'best-action queries', which are queries that informs the algorithm the best option at the current time step. Prior work considers different form of predictions or queries, such as a vector correlated with the actual loss function, or the best option amongst a small subset of all the options. This work differs from prior works both in the form of the advice, and that the learning algorithm is only allowed to query at $k$ out of $T$ time steps.

WIthin this advice model, the authors show that in either the full feedback model, in which the full loss function is revealed to the algorithm after every time step, or the label efficient feedback model, in which the algorithm is only given the full loss function if it issues a query in the current round, the query budget $k$ multiplicatively affects (decreases) regret. This is a surprising result since the regret per round is upper bounded by $1$, so $k$ rounds can only increment the regret additively instead of multiplicatively. The algorithm the authors propose is a very simple and intuitive modification of the classical Hedge, or multiplicative weights, algorithm, except that in $k$ out of $T$ rounds, randomly sampled according to some procedure, the algorithm queries the best-action and follows it instead of a weighted selection amongst all options.

**Strengths:**

The paper studies the intersection between two interesting fields, learning-augmented algorithms and online learning, and contributes positively to its literature. Traditionally, the field of learning-augmented algorithms places a focus on (classical) online algorithms, so whether external predictions can help online learning is a very natural extensional question to ask. Over prior works in the same intersection, the authors additionally showcase the power of such predictions by showing that a small number of queries or predictions can  significantly decrease the loss/regret of online learning algorithms, which is a very surprising and exciting discovery.

**Weaknesses:**

I find the author's presentation somewhat rushed and unsatisfactory, and makes significant assumptions on the reader's background knowledge. Coming from a learning-augmented algorithm background, I find many online learning aspects of the paper not sufficiently defined or explained in a self-contained way, despite having some familiarity with the field. Some minor points of complaint are listed in the Questions section below.

More importantly, a majority of the main corpus of the paper consists of technical proofs following one another, without much introduction or intuition in between to give the readers a high-level guideline of what is going on in the paper. It is very hard, without full familiarity of relevant literature, to follow the author's proofs, and understand what part a claim or statement plays in the grand scheme of proving the main theorems. As a result, while I believe the author's claims are overall correct, I am unable to confidently confirm the technical soundness of the paper.

Overall I still believe that this is an exciting paper; but its presentation can be improved significantly.

**Questions:**

The authors discuss the potential of using noisy, possibly erroneous predictions or queries in the conclusion. From my understanding, the learning-augmented algorithms community put a lot of emphasis on utilizing machine-learned oracles that can possibly be accurate but have no rigorous guarantees to facilitate and *augment* classical algorithms to retain both *consistency* (good performance when predictions are accurate) and *robustness* (good performance even when predictions are arbitrarily bad). As a result, I would be very interested in seeing discussions and extensions in this noisy prediction regime in the future.

As illustrated above, I believe that a lot of intuition, such as high-level description of the proof strategy, and the usage and purpose of each component lemma or claim in colloquial terms, would be very helpful to the overall presentation in the main corpus, and can help guide readers to follow the flow of logic.

Some minor details I would like to point out:
- Line 35-36: "... the platform's task **consists in** deciding whether..." should be "consists of"?
- Line 49: "...with full feedback, **an** in the **stochastich** multi-armed...".
- The phrases "loss" and "feedback" are both used a lot, perhaps interchangeably, which is somewhat confusing. This is especially important in the label-efficient feedback section, in which it is not immediately clear what is "partial" about the "partial feedback" given to the algorithm. Is it just that the algorithm receives the loss function only when it makes a query, or is the received loss function itself partial?
- Line 102-103: The entire sentence within the parenthesis does not make sense to me, grammatically.
- Line 120: The introduction contains multiple mentions of the classical Hedge algorithm, but never explains, in any level of detail, what is the Hedge algorithm. From their pseudocode of Algorithm 1, which is described as a modification of Hedge, I can extrapolate that Hedge is the classical multiplicative weight algorithm, but the authors did not even make this clear in the introduction.
- Line 142: "We have the following theorem." Which theorem is this sentence exactly referencing to? Theorem 2.2 is the natural candidate but its statement predates this sentence.
- Line 162: I find it somewhat slightly strange that of the two algorithms presented in the paper, one is given a pseudocode block, while the other is only given a description.
- Line 196: "...we construct two **of** randomized instances...".
- Line 199-200: "(and $Z_t$ is an empty $n$-dimensional vector)". Is the scope of this sentence if a query is not issued at time $t$?

**Limitations:**

The authors sufficiently discuss limitations and future directions in the conclusion section. There are no ethical concerns or limitations in their work.

---

> ### Author Rebuttal · Authors · 2024-08-05
>
> We thank the reviewer for the questions and suggestions, which we will implement in the final version of the paper.
>
> **Question.** The authors discuss the potential of using noisy ... prediction regime in the future.
>
> **Answer.**  We thank the reviewer for this suggestion. In several applications, these external queries come from human reviewers, for whom we can assume that query responses are perfect (see, e.g., https://transparency.meta.com/en-gb/policies/improving/content-actioned-metric/, https://transparency.meta.com/en-gb/policies/improving/prioritizing-content-review/, https://support.google.com/adspolicy/answer/13584894). However, if the queries are generated by external learning algorithms (even if trained on this specific task), they may be erroneous. The suggested direction is an interesting open questions, which we defer to future work. We believe that our paper's techniques for perfect best-action queries will be essential to analyze these model's extensions.
>
> **Question (cont'd).** As illustrated above, I believe ... follow the flow of logic.
>
> **Answer.** We appreciate the reviewer’s feedback. We tried to add an overarching explanation of the proof strategy in Section 1.3. We detailed the technical challenges and how we addressed them, providing an explicit explanation of how the various components fit together and highlighting the advantages of uniform querying. We will expand the explanation in the technical sections, exploiting the extra page that is provided for the final version of the paper.
>
> **Minor details.**
>
> - Q - Line 49: "...with full feedback, an in the stochastich multi-armed...". The phrases "loss" and "feedback" are both used a lot, perhaps interchangeably, which is somewhat confusing. This is especially important in the label-efficient feedback section, in which it is not immediately clear what is "partial" about the "partial feedback" given to the algorithm. Is it just that the algorithm receives the loss function only when it makes a query, or is the received loss function itself partial?
>
> - A - Yes, the algorithm receives the \textit{entire} loss vector when it makes the query, and before choosing the action. So, the loss vector itself is given as a whole (and, thus, not partial) but only when a query is issued.
>
> - Q - Line 120: The introduction contains multiple mentions of the classical Hedge algorithm, but never explains, in any level of detail, what is the Hedge algorithm. From their pseudocode of Algorithm 1, which is described as a modification of Hedge, I can extrapolate that Hedge is the classical multiplicative weight algorithm, but the authors did not even make this clear in the introduction.
>
> - A - Yes, it indeed is the classical Hedge algorithm. However, for the purposes of an easier analysis and calculations, we subtract $\ell_t(i^*_t)$ in the update rule exponent. As the reviewer points out, however, the distribution from which we sample the next action is the same as in classical Hedge (the exponent cancels out in numerator vs. denominator).
>
> - Q - Line 142: "We have the following theorem." Which theorem is this sentence exactly referencing to? Theorem 2.2 is the natural candidate but its statement predates this sentence.
>
> - A - We actually meant "We have the following algorithm." Thank you for pointing this out.
>
> - Q - Line 199-200: ``(and is an empty $n$-dimensional vector)''. Is the scope of this sentence if a query is not issued at time?
>
> - A - Thank you for pointing this out. We meant to write ``(and is an empty $n$-dimensional vector otherwise)'', i.e., we do not observe anything if we do not query. We will edit it in the final version of the paper.

---

> > ### Comment · Reviewer_tNck · 2024-08-08
> >
> > Thank you for your response. I maintain my rating. Overall I agree with the other reviewers that the paper asks a very timely and interesting question about whether we can utilize reliable queries (and hopefully in the future, learning-augmentation with unreliable predictions) to facilitate online learning, and I very much hope to see the paper structured better with more intuition and self-contained proofs, and suggestions from me and other reviewers incorporated, in the camera-ready version, if accepted.

---

### Official Review · Reviewer_VoCp · 2024-07-09

**Soundness:** 3
**Presentation:** 2
**Contribution:** 3
**Rating:** 6
**Confidence:** 3

**Summary:**

The paper’s title concisely summarizes the topic. They study both the “full feedback” setting, where the learner observes the entire loss vector (what the loss would have been for each possible action) on each time step, and the “label-efficient feedback” setting, where the learner only observes the loss vector on time steps when a query is issued.

The results are as follows:
1. An algorithm for the full feedback setting which achieves regret $O(\min(\sqrt{T}, T/k))$ with k best action queries. Interestingly, the queries are issued uniformly at random.
2. An algorithm for the label-efficient feedback setting which achieves regret $O(\min(T/\sqrt{k}, T^2/k^2))$. The queries are also issued mostly at random here (specifically, uniformly at random until the budget is exhausted).
3. Lower bounds for both of the above results which are asymptotically tight.

**Strengths:**

I find the model of sublinear best-action queries to be quite natural, and the authors also do a good job of motivating it. One example given in the paper which I found compelling was an algorithm for flagging harmful content which can escalate to a human review a limited number of times.

I also found the results to be impressive. As the authors state,

> Note, the total of the losses incurred by any algorithm in k rounds is at most k, which only affects the overall regret in an additive way; nevertheless, we prove that issuing k queries has an impact on the regret that is multiplicative in k.

This seems like a potentially powerful insight to me.

**Weaknesses:**

I found the paper to be quite technically dense. Although the description of the results in the introduction was clear, the authors don’t provide much intuition for their results. I had a hard time identifying where (on a technical level) the multiplicative power of k comes from. Conveying technical intuition along with technical results is very beneficial for other researchers trying to make use of the insights from this paper. I suspect the impact of the current version of paper is limited by the presentation.

For transparency, I will mention that I am not an expert in this sub-area, and it is possible that experts would not have the same complaint. However, to the extent that the intended audience of this paper is researchers in related but not identical fields, I think this weakness is significant.

See also the “limitations” section below.

**Questions:**

Can you provide technical intuition for where the multiplicative power of the best-action comes from?

Below are minor comments that the authors are free to use or discard. I’m including them in this section because I don’t know where else to put them, but I don’t expect responses.
-  Line 71: “In our paper we pinpoint the exact minimax regret rates for the problems studied.” It might be worth saying “asymptotically tight” instead of “exact” (although it’s pretty clear from context).
- Lines 84 - 86 are quite compelling.
- Line 93: Maybe mention that the learner still observes the entire loss vector, not just the best action.
- Lines 132 - 133: “combining the Hedge algorithm [Chapter 2 in Cesa-Bianchi and Lugosi, 133 2006] to uniform queries” → “with uniform queries”?
- Line 200: “and Z_t is an empty n-dimensional vector” → “and Z_t is an empty n-dimensional vector otherwise”?

**Limitations:**

In Question 2 of the checklist, the authors state:
> The paper has no significant limitation. Being theoretical in nature, there is a set of model assumptions

I found this answer to be disappointing. Theoretical papers absolutely have limitations. Are the model assumptions realistic? If not, to what extent do the results crucially hinge on those assumptions? Do the algorithms have inefficiencies that would impede or preclude practical usage? Are the technical insights too complex to be easily understandable by other researchers? These are just example limitations; I’m not claiming that any of these particular examples apply to this paper.

---

> ### Author Rebuttal · Authors · 2024-08-05
>
> We thank the reviewer for the questions and suggestions, which we will implement in the final version of the paper.
>
> **Question 1.** Can you provide technical intuition for where the multiplicative power of the best-action comes from?
>
> **Answer.** We are happy that the reviewer shares our enthusiasm for the (somewhat) surprising effectiveness of our query model. We provide here two high-level observations that may help a better understanding. We will add them to the final version of the paper:
>
> - The additive term $\frac{k}{T} \cdot L_T^{\min}$ (given in Observation 2.3, and in particular right after line 150) impacts the choice of the learning rate, which is allowed to be more aggressive, thus impacting the regret in a multiplicative way. To be more specific, in the usual Hedge performance analysis, we do not care about the negative term $\frac{-\eta L_T^{\min}}{1-\eta}$ (in the inequality given right after line 153). This negative term together with the additive $\frac{k}{T} \cdot L_T^{\min}$ term given by best-action queries allows us to set the optimal $\eta$ to be larger than the usual (order of) $1/\sqrt{T}$. In other words, the additive impact of the $\frac{k}{T} \cdot L_T^{\min}$ term permits a multiplicative gain in regret as the learning rate $\eta$ is modified and increased.
>
> - Consider the following natural instance, which provides a simple proof of the $\Omega(\sqrt{T})$ regret lower bound in the adversarial setting (without queries). The instance is composed of two arms, whose rewards are i.i.d. Bernoulli distribution with probability $0.5$. Any learning algorithm achieves $T/2$ regret, while the best-fixed arm in hindsight is expected to achieve an extra $\Theta(\sqrt T)$ term (this is a simple corollary of the expected distance of a random walk). Now, if the learner is given the power to issue $\approx \sqrt{T}$ queries, then its regret naturally drops from $\sqrt{T}$ to constant.
>
> **Limitation 1.** Are the model assumptions realistic? If not, to what extent do the results crucially hinge on those assumptions?
>
> **Answer.** We thank the reviewer for pointing this out. The model is realistic as the number of queries we can issue is limited. Moreover, if we supposed that the external queries come from a human (which is true in many applicative settings), we could assume that these reviews are noiseless (see, e.g., https://transparency.meta.com/en-gb/policies/improving/content-actioned-metric/, https://transparency.meta.com/en-gb/policies/improving/prioritizing-content-review/, https://support.google.com/adspolicy/answer/13584894). This, of course, no longer holds when queries are generated by an external learning algorithm, where some degree of noise is inevitable. Our current model does not account for potentially faulty queries. Indeed, our algorithms ``blindly trust'' the query, and the bounds presented in our analysis rely on the assumption that the query is perfect. For example, the former intuitive explanation of the multiplicative power given by best-action queries no longer holds. However, we believe that our work, which provides a nearly complete understanding of the perfect prediction case, may work as a natural starting point along such direction. We will expand on this in the final version of the paper, and we will update the checklist accordingly.

---

> > ### Comment · Reviewer_VoCp · 2024-08-10
> >
> > Thanks for the response. I am satisfied and am maintaining my positive rating.

---

### Official Review · Reviewer_aUFF · 2024-07-16

**Soundness:** 4
**Presentation:** 3
**Contribution:** 2
**Rating:** 5
**Confidence:** 3

**Summary:**

This paper considers an online learning with actions model where the learner is allowed to make $k$ "best action queries". Such a query at step $t \in [T]$ will return $i_t^\ast \in \arg\min_{i \in [n]} \ell_t(i)$; an action that minimizes the loss at step $t$. The loss values are bounded (in $[0, 1]$ w.l.o.g) and may be generated by an oblivious adversary. The best action query is performed _before_ the learner makes the decision (arm choice). The regret of the learming algorithm is defined in the usual way.

They first consider the full feedback setting, where _after_ making the arm choice, the learner observes the entire loss vector $(\ell_t(1),\ldots,\ell_t(n))$. In this setting, without best action queries, the Hedge algorithm gives $\tilde{O}(\sqrt{T})$ regret for bounded losses (and this is well-known to be optimal).

(i) In the full feedback setting (augmented with best action queries), the authors show that the Hedge algorithm with $k$ uniformly random best action queries gives an (expected)  regret of $\tilde{O}\left(min\\{\sqrt{T},T/k\\}\right)$. This beats the standard regret bound when $k$ is $\omega(\sqrt{T})$.

(ii) They also show a matching lower bound (up to $\log n$ factors) that any full-feedback algorithm with $k$ best action queries (not necessarily random) will suffer $\Omega(\sqrt{T},T/k)$ (expected) regret.

They then consider a label-efficient feedback model w.r.t the best action queries. Here the restriction in feedback is entirely in terms of observing the loss values (there is no outcome space) and linked to the best action queries themselves. The learner can make $k$ best action queries; if the learner chooses to make a query at time step $t$, it receives the full feedback after it makes the choice. Otherwise, it receives no feedback.

(i) In the label-efficient feedback setting (augmented with best action queries), the authors show that the label-efficient Hedge algorithm (does not update the probabilities when there is no feedback) with random best action queries ($\leq k$ of them, but not quite uniform random unlike in full feedback) achieves $O\left(\min(T/\sqrt{k},T^2/k^2)\right)$ regret. This beats the known regret bound for label-efficient Hedge (from Cesa-Bianchi and Lugosi 2006) when $k$ is $\omega(T^{2/3})$.

(ii) They also show a matching lower bound for regret (up to $\log n$ factors) of any algorithm with $k$ query label efficient feedback.

In the appendix, they show nearly identical bounds for full feedback and label-efficient feedback in the stochastic losses setting. Here the algorithms are Follow the leader and Explore then commit, and the best action queries are simply made in the first $k$ rounds.

**Strengths:**

* The model (best action queries) is theoretically interesting and the results are quite nice in the sense that you can get polynomial improvements in the regret by incorporating _sublinearly-many random_ best action queries with the standard (full feedback and label efficient feedback) Hedge algorithms.
* The paper is well-written. Almost all the proofs are cleanly presented in the main matter itself. They also survey and compare some of the existing work on augmenting online prediction algorithms with additional information.

**Weaknesses:**

* Even though the best action query model yields theoretically interesting results, it is a bit too strong compared to some of the existing work. Even if only sublinearly many queries are made, each query is supposed to give the _correct best action (among all)_ for that time step before making the decision, and the number of queries needed is still polynomial in $T$ (at least $T^{1/2}$) to get any interesting result w.r.t the regret (in both full feedback and label efficient feedback). When the authors link best action queries to human expert advice in the introduction section, the assumption on the queries becomes somewhat impracticable.

**Questions:**

* In the equations after line 178, is the last equality an inequality ($\leq$) using $T\eta \leq \hat{k}$?

**Limitations:**

* There is no negative social impact.

* The major practical limitation is the strong requirements of the query model. It would be much more feasible if (i) the queries returned an "approximate" best action among all, (ii) the queries returned the exact best action among a subset of actions or (iii) interesting results were obtainable with logarithmically many queries.

---

> ### Author Rebuttal · Authors · 2024-08-05
>
> We thank the reviewer for the questions and suggestions, which we will implement in the final version of the paper.
>
> **Question.** In the equations after line 178, is the last equality an inequality using $T\eta \leq \hat k$?
>
> **Answer.** Yes, thank you for pointing this out.
>
> **Limitation.** The major practical limitation is the strong requirements of the query model. It would be much more feasible if (i) the queries returned an "approximate" best action among all, (ii) the queries returned the exact best action among a subset of actions or (iii) interesting results were obtainable with logarithmically many queries.
>
> **Answer.** We thank the reviewer for this suggestion. In several applications (see, e.g., https://transparency.meta.com/en-gb/policies/improving/content-actioned-metric/, https://transparency.meta.com/en-gb/policies/improving/prioritizing-content-review/, https://support.google.com/adspolicy/answer/13584894), these external queries come from human reviewers, for whom we can assume that query responses are perfect.  We agree that in some applications, i.e., when the queries are generated by external learning algorithms, some degree of noise may be inevitable. We believe that addressing also these applications is a compelling and interesting direction of research, and we believe that our work, which provides a nearly complete understanding of the perfect prediction case, may work as a natural starting point along such direction.

---

### Official Review · Reviewer_Sy4T · 2024-07-21

**Soundness:** 4
**Presentation:** 4
**Contribution:** 3
**Rating:** 7
**Confidence:** 4

**Summary:**

This paper considers the standard prediction with expert advice setting of online learning, but with the twist that the learner may issue, up to $k$ times, a "best-action query" before making a prediction, in which case the identity of an expert incurring the smallest loss in the round is revealed to the learner (and of course they can choose that expert on that round). Two settings are investigated: the standard experts setup wherein the losses of all experts are revealed after each round, and the 'label-efficient prediction' (LEP) setting, where the losses are not revealed unless the learner issues a query for the same (which the authors identify with the best-action query, i.e., all losses are only revealed when the learner makes a best-action query).

The paper shows that these $k \ll T$ best-action queries have an impressive effect, improving the regret in the experts setting to $O(\min(\sqrt{T}, T/k)$, and in the LEP setting to $O(\min(T/\sqrt{k}, T^2/k^2)$. Surprisingly, the method achieving this is just hedge, but run with losses of the form $\ell_{t,i} - \min_i \ell_{t,i}$ instead of just $\ell_{t,i}$ (and the appropriate modification a la prior work on LEP for this setting). The approach to showing this uses the standard analysis of the hedge algorithm, and uses the additive advantage of the best-action queries to allow the method to set an aggressively large learning rate (when $k$ is large enough) without suffering a strong penalty for the same, leading to improved regret bounds. The paper concludes by showing matching lower bounds for this setup, in a minimax sense.

**Strengths:**

I think that this paper is both pertinent and timely: the investigation of how augmented information can improve regret in classical online learning settings has been an interesting direction of research in the recent years, and the paper makes a significant contribution to this body of work. I particularly find the improvement in the LEP setting to be remarkable. The paper is further well written and easy to understand, with existing arguments being used in an interesting new way.

**Weaknesses:**

I don't see major weaknesses in the paper.  I think the only change that is necessary is that the discussion of the related work in section C should be moved to the main paper, since this provides valuable context to the investigation presented within (but this should be easily accommodated with the extra page, if the paper is accepted).

Perhaps another point of improvement lies in deepening the discussion of scenarios where the model being studied has pertinence. The moderation setup certainly is interesting and natural, but are there other situations where the authors see the relevance of best-action queries?

**Questions:**

Suggestion: One approach to modeling problems like using limited moderation effectively that I have seen is through the abstention model of learning, wherein the learner may "abstain" on a query, and receive a "best-response" by utilising extra resources. I think work on online abstention may thus be a useful point of contact with the literature that the paper misses, and might be worth including a discussion on. This includes both the KWIK model, and the full-information model. Useful points of contact with this literature are below.

Li, Littman, Walsh, and Strehl, Machine Learning 2011, Sayedi, Zadimoghaddam, and Blum NIPS10; Zhang and Chowdhuri, COLT 16; Cortes et al., ICML 18; Neu and Zhivotovskiy, COLT 20; Gangrade et al., Neurips 21

**Limitations:**

This is fine

---

> ### Author Rebuttal · Authors · 2024-08-05
>
> We thank the reviewer for the questions and suggestions, which we will implement in the final version of the paper.
>
> **Weakness.** Perhaps another point of improvement ... best-action queries?
>
> **Answer.** One other application of our model is in fraud detection. In this context, an online learning algorithm aims to identify incoming points as either potential frauds or benign content. This is critically important for platforms like Booking.com, where customers defrauded by fictitious hotels or BnBs must be reimbursed. Proactively identifying potential fraud saves the platform substantial amounts of money. Specifically, Tax, Jan de Vries, de Jong, Dosoula, van den Akker, Smith, Thuong, and Bernardi (MLHat, 2021) provide evidence of a machine learning system designed to detect potentially fraudulent listings on Booking.com. This system is supplemented by expert human oversight, which can be called upon a limited number of times. These instances of expert involvement can be interpreted as best-action queries.
>
> **Suggestion.** One approach to modeling problems ... Gangrade et al., Neurips 21.
>
> **Answer.** We thank the reviewer for providing this point of contact. We will make sure to expand upon this connection in the final version of the paper related work.

---

### Decision · Program_Chairs · 2024-09-25

**Decision:**

Accept (poster)

**Comment:**

All the referees are unanimous in the fact that this paper contributes a new model for online learning where best-action feedback can be elicited at a cost. The model is well-motivated, the analysis of the proposed algorithm shows surprisingly strong results, and the paper is overall very well written. I am glad to accept it for presentation at Neurips.